**Assessing MOPITT carbon monoxide retrievals over urban versus non-urban regions**
**Wenfu Tang[1,2], Helen M. Worden[2], Merritt N. Deeter[2], David P. Edwards[2], Louisa K.**
**Emmons[2], Sara Martínez-Alonso[2], Benjamin Gaubert[2], Rebecca R. Buchholz[2], Glenn S.**
**Diskin[3], Russell R. Dickerson[4], Xinrong Ren[4,5], Hao He[4], and Yutaka Kondo[6]**
[1]Advanced Study Program, National Center for Atmospheric Research, Boulder, CO, USA
[2]Atmospheric Chemistry Observations and Modeling, National Center for Atmospheric Research,
Boulder, CO, USA
[3]NASA Langley Research Center, Hampton, VA, USA
[4]Department of Atmospheric and Oceanic Science, University of Maryland, College Park, MD,
USA
[5]Air Resources Laboratory, National Oceanic and Atmospheric Administration, College Park, MD,
USA
[6]National Institute of Polar Research, Tachikawa, Japan
*Correspondence: Wenfu Tang (wenfut@ucar.edu)*
**Abstract**

17       The Measurements of Pollution in the Troposphere (MOPITT) retrievals over urban

regions have not been validated systematically, even though MOPITT observations are widely
used to study CO over urban regions. Here we compare MOPITT products over urban and non-
urban regions with aircraft measurements from DISCOVER-AQ (2011-2014), SEAC[4]RS (2013),
ARIAs (2016), A-FORCE (2009; 2013), and KORUS-AQ (2016) campaigns. In general, MOPITT
agrees reasonably well with the in-situ profiles, over both urban and non-urban regions.  Version
8 multispectral product (V8J) biases vary from -0.7% to 0.0% and version 8 thermal-infrared
product (TIR) biases vary from 2.0% to 3.5%. The evaluation statistics of MOPITT V8J and V8T
over non-urban regions are better than that over urban regions with smaller biases and higher
correlation coefficients. We find that the agreement of MOPITT V8J and V8T with aircraft
measurements at high CO concentrations is not as good as that at low CO concentrations, although
CO variability may tend to exaggerate retrieval biases in heavily-polluted scenes. We test the
sensitivities of the agreements between MOPITT and in-situ profiles to assumptions and data
filters applied during the comparisons of MOPITT retrievals and in-situ profiles. The results at the
surface layer are insensitive to the model-based profile extension (required due to aircraft altitude
limitations) whereas the results at levels with limited aircraft observations (e.g., the 600-hPa layer)
are more sensitive to the model-based profile extension. The results are insensitive to the maximum
allowed time difference criterion for co-location (12 hours, 6 hours, 3 hours, and 1 hour), and are
generally insensitive to the radius for co-location, except for the case where the radius is small (25
km) and hence few MOPITT retrievals are included in the comparison. Daytime MOPITT products
have smaller overall biases than nighttime MOPITT products when comparing both MOPITT
daytime and nighttime retrievals to the daytime aircraft observations. However, it would be
premature to draw conclusions on the performance of MOPITT nighttime retrievals without
nighttime aircraft observations. Applying signal-to-noise ratio (SNR) filters does not necessarily
improve the overall agreement between MOPITT retrievals and in-situ profiles, likely due to the
reduced number of MOPITT retrievals for comparison. Comparisons of MOPITT retrievals and
in-situ profiles over complex urban or polluted regimes are inherently challenging due to spatial
and temporal variabilities of CO within MOPITT retrieval pixels (i.e., footprints). We demonstrate
that some of the errors are due to CO representativeness with these sensitivity tests, but further
quantification of representativeness errors due to CO variability within the MOPITT footprint will
require future work.

## 49    1. Introduction

Observations from the Measurements of Pollution in the Troposphere (MOPITT)

instrument onboard the NASA Terra satellite have been used for retrieving total column amounts
and volume mixing ratio (VMR) profiles of carbon monoxide (CO) using both thermal-infrared
(TIR) and near-infrared (NIR) measurements since March 2000. Besides the TIR-only and NIR-
only products, the MOPITT multispectral TIR-NIR product is also provided, which has enhanced
the sensitivity to near-surface CO (Deeter et al., 2011, 2013; Worden et al., 2010). Since the start
of the mission, the MOPITT CO retrieval algorithm has been improved and enhanced continuously
(Worden et al., 2014). For example, the Version 6 product improvements included the reduction
of both a geolocation bias and a significant latitude-dependent retrieval bias in the upper
troposphere (Deeter et al., 2014). In the Version 7 products, a new strategy for radiance-bias
correction and an improved method for calibrating MOPITT's NIR radiances were included
(Deeter et al., 2017). For the most recently released MOPITT Version 8 products, enhancements
include a new radiance bias correction method (Deeter et al., 2019). Meanwhile, the MOPITT
products have been extensively evaluated and validated with in-situ measurements, though this
has been done primarily over non-urban areas (Deeter et al., 2010, 2012, 2013, 2014, 2016, 2017,
2019; Emmons et al., 2004, 2007, 2009). In addition, MOPITT products have also been compared
with ground-based spectrometric column retrievals (e.g., Buchholz et al., 2017; Hedelius et al.,
2019). For the past two decades, MOPITT CO products have been widely used for various
applications, including understanding atmospheric composition, evaluating atmospheric chemistry
models, and constraining inverse analyses of CO emissions (e.g., Arellano et al., 2004, 2006, 2007;
Chen et al., 2009; Edwards et al., 2006; Emmons et al., 2010; Fortems-Cheiney et al., 2011;
Gaubert et al., 2016; Heald et al., 2004; Jiang et al., 2018; Kopacz et al., 2009, 2010; Kumar et al.,
2012; Lamarque et al., 2012; Tang et al., 2018; Yurganov et al., 2005).
MOPITT products are particularly useful for monitoring and analyzing air pollution over
urban regions because of the enhanced retrieval sensitivity to near-surface CO and the long-term
record (e.g., Clerbaux et al., 2008; Girach and Nair, 2014; Jiang et al., 2015, 2018; Kar et al., 2010;
Tang et al., 2019; Worden et al., 2010; Li and Liu, 2011; He et al., 2013; Aliyu and Botai, 2018;
Kanakidou et al., 2011). However, the performance of MOPITT retrievals over urban regions has
not yet been validated systematically. Furthermore, in-situ observations of CO profiles over urban
areas are limited, especially in Asia. Indeed, along with the non-urban validation exercises
mentioned above, development and validation of the MOPITT retrieval algorithm relies heavily
on in-situ measurements over remote regions, such as measurements from the HIAPER Pole-to-
Pole Observations (HIPPO) and the Atmospheric Tomography Mission (ATom) campaigns (e.g.,
Deeter et al., 2013, 2014, 2017, 2019). Comparisons of MOPITT products to measurements with
aircraft profiles during the Korea United States Air Quality (KORUS-AQ) campaign over South
Korea have only recently been made in Deeter et al. (2019), but without explicitly analyzing
MOPITT performance over urban regions.
In this study, we compare MOPITT Version 8 and 7 products with aircraft profiles made
over urban regions (as well as non-urban regions) from campaigns including: Deriving Information
on Surface conditions from Column and Vertically Resolved Observations Relevant to Air Quality
(DISCOVER-AQ); the Studies of Emissions and Atmospheric Composition, Clouds, and Climate
Coupling by Regional Surveys (SEAC[4]RS); the Air Chemistry Research In Asia (ARIAs); the
Aerosol Radiative Forcing in East Asia (A-FORCE); and KORUS-AQ. These campaigns are
described in Section 2, along with a brief description of the MOPITT products and the
methodology used. We present the comparisons of MOPITT products to aircraft profiles, and
discuss the impacts of key factors in the retrieval process on the retrieval results in Section 3. In
Section 4, we discuss the sensitivities of results to the assumptions and data filters made for
aircraft-satellite comparisons not only in this study, but also in previous evaluation studies of
MOPITT and other satellite products. Section 5 gives the conclusions of the study.

**2. Data and methods**
**2.1 MOPITT retrievals and products**

MOPITT is a nadir sounding satellite instrument flying on the NASA Terra satellite. It uses

a gas filter correlation radiometer and measures radiance at both the TIR band near 4.7 μm and the
NIR band near 2.3 μm. These observations have a spatial resolution of about 22 km × 22 km with
satellite overpass time at approximately 10:30 and 22:30 (local time). To determine a unique CO
concentration profile from the MOPITT measured radiances, an optimal estimation-based retrieval
algorithm, and a fast radiative transfer model are used (Deeter et al., 2003; Edwards et al., 1999).
The retrieved state vector ($x_{rtv}$) for optimal estimation-based retrievals can be expressed as

$$x_{rtv} = x_a + A(x_{true} - x_a) + \epsilon \qquad (1)$$

$x_a$ and $x_{true}$ are the a priori state vector and the true state vector, respectively. $A$ (which has a size
of 10×10) is the retrieval averaging kernel matrix (AK) that represents the sensitivity of retrieved
profiles to actual profiles and $\epsilon$ is the random error vector. Note that CO quantities in the state
vector are retrieved as $\log_{10}$(VMR).

We focus on validating the recently released Version 8 of the MOPITT TIR, NIR, and

multispectral TIR-NIR products. We also include comparisons with the MOPITT Version 7 TIR,
NIR, and multispectral TIR-NIR products in the Section 3.1 for reference. These two versions of
MOPITT products were introduced in detail in Deeter et al. (2017) and Deeter et al. (2019).

## 2.2 Aircraft measurements used for comparisons

Aircraft-sampled profiles of CO concentrations during the DISCOVER-AQ, SEAC[4]RS, ARIAs, A-FORCE, and KORUS-AQ campaigns are used for comparisons with MOPITT-retrieved profiles. DISCOVER-AQ, and SEAC[4]RS were conducted over the US, while ARIAs, A-FORCE, and KORUS-AQ were conducted over East Asia. Locations of the aircraft profiles from these campaigns are compared with the MODIS (Moderate Resolution Imaging Spectroradiometer) Terra+Aqua Land Cover Type Climate Modeling Grid Yearly Level 3 Version 6 0.05°×0.05° Global product (MCD12C1 v006) (Friedl and Sulla-Menashe, 2015) to determine if a profile was sampled over an urban or non-urban region. Specifically, for each aircraft profile, a 0.5°×0.5° box centered over the location of the aircraft profile (determined by averaged latitude and longitude of aircraft observations in the profile) is selected. If the urban and built-up fraction in the box is larger than 10%, the profile is considered to be an urban profile. Overall, for each campaign, the averaged aircraft profile over urban regions has higher CO concentrations compared to that over non-urban regions, especially near the surface (see Figure S1). Profiles during ARIAs, which are sampled over Hebei province in China, are exceptional, as the averaged profile over non-urban regions has higher CO concentrations especially near the surface, indicating high CO levels in the entire study region. We note that Hebei is one of the most heavily industrialized and polluted regions, and the difference in CO profiles is driven less by urban versus rural than by synoptic and mesoscale meteorology. In addition, Hebei is an arid region and subject to strong nocturnal inversions, so the surface CO can be very high. For aircraft profiles sampled during KORUS-AQ, the CO profiles over urban and non-urban regions are similar, even though the averaged profile over urban regions has slightly higher CO concentration near the surface. This is largely due to the fact that many of the non-urban aircraft profiles are sampled over the Taehwa forest site, which is impacted by CO transported from the nearby Seoul urban region. The urban regions often have different surface parameters (e.g., surface temperature and emissivity), and usually but not always have higher CO concentrations than non-urban regions. However, the surface parameters are unlikely to impact the ultimate quality of MOPITT retrieval products (Pan et al., 1998; Ho et al., 2005). The goal of this study is to understand if MOPITT retrievals are able to represent conditions over urban regions given sampling, and cloud cover. In addition, the relatively large spatial and temporal variability of CO concentrations over urban regions makes the validation even more complex. Because of the complexity of urban regions and their connection with non-urban regions nearby, we also provide

analysis at high CO concentrations regardless of landcover type. As the reviewer pointed out, the
comparisons are done for the 600-hPa layer (usually in the free troposphere). It is possible that CO
concentrations at this layer are transported from other regions that are not representative of urban
regions. Even so, MOPITT retrievals at the 600-hPa layer are still impacted by the CO
concentrations at other layers including the surface layer (equation 1). Therefore, the comparisons
at 600 hPa is necessary.
The campaigns and profiles are summarized in the Table 1 and Figure 1. During
DISCOVER-AQ, SEAC[4]RS, and KORUS-AQ, CO concentrations were measured by the NASA
Differential Absorption Carbon monOxide Measurement (DACOM), whereas during ARIAs and
A-FORCE, CO concentrations were measured by Picarro G2401-m and Aero-Laser GmbH
AL5002, respectively. Note that the primary goal of DISCOVER-AQ was to provide aircraft
observation methodologies for satellite validation (e.g., Lamsal et al. (2014)). There are 121
profiles over four urban regions from DISCOVER-AQ, making it particularly useful for the goal
of this study. Because of this, our results are heavily driven by aircraft profiles from DISCOVER-
AQ. Even though there are only two profiles sampled over urban regions, the A-FORCE campaign
obtained 45 profiles in total sampled over East Asia during Spring 2009, Winter 2013, and Summer
2013. The seasonal and spatial coverage of the dataset makes it representative of the region. The
ARIAs campaign provides 19 profiles and three of these were sampled over Chinese urban regions.
Few previous studies have validated MOPITT products over China (e.g., Hedelius et al., 2019), so
aircraft profiles from ARIAs have also been included in this study.
**2.3 Method for comparing MOPITT profiles to aircraft measurements**
We generally follow the method that has been used in previous MOPITT evaluation and
validation studies (Deeter et al., 2010, 2012, 2013, 2014, 2016, 2017, 2019; Emmons et al., 2004,
2007, 2009). There are four main steps in aircraft versus MOPITT comparisons.
(1) Because of aircraft altitude limitations, in-situ data from field campaigns do not typically reach
the highest altitudes at which MOPITT radiances are sensitive.  Therefore, to obtain a complete
vertical profile as required for comparison with MOPITT retrievals, each in-situ profile is extended
vertically using the following steps: (i) the aircraft measurements are interpolated to the 35-level
vertical grid used in MOPITT forward model calculations (0.2–1060 hPa); (ii) the levels from the
surface to the lowest-altitude aircraft measurement are filled with the value of the in-situ
measurement at the lowest-altitude aircraft measurement; (iii) for levels above a certain pressure
level $P_{interp}$ (higher altitude), model or reanalysis data are used directly; (iv) for levels between the
highest-altitude aircraft measurement and the altitude of $P_{interp}$, values are linearly interpolated.
Unlike the previous MOPITT evaluation studies that used monthly model results from MOZART
(Model for OZone And Related chemical Tracers) (Emmons et al., 2010) or CAM-chem
(Community Atmosphere Model with chemistry) (Lamarque et al., 2012), here we use 3-hourly
Copernicus Atmosphere Monitoring Service (CAMS) reanalysis of CO produced by the European
Centre for Medium-Range Weather Forecasts (ECMWF). CAMS CO reanalysis has a horizontal
resolution of 80 km × 80 km, and 60 vertical grids (from surface to 0.1 hPa). Satellite retrievals of
atmospheric composition including MOPITT TIR Version 6 total column CO retrievals are
assimilated       in       the       CAMS       reanalysis       (Inness       et       al.,       2019;
https://confluence.ecmwf.int/pages/viewpage.action?pageId=83396018). We note that as we do
not compare with these higher levels later, the use of CAMS reanalysis is expected to have a
minimal impact on the lower levels we use in the comparison (e.g., the surface layer, the 800-hPa
layer, and the 600-hPa layer). The final CO profile at the 35-level vertical grid is then regridded
onto a coarser 10-level grid (for consistency with the actual MOPITT retrieval grid) by unweighted
averaging the fine-grid VMR values in the layers immediately above the corresponding levels in
the retrieval grid. We investigate the sensitivity of the results to $P_{interp}$ in Section 4.1.
(2) For a given in-situ profile, MOPITT profiles are considered co-located with the aircraft profile
and are selected for comparison only if their center points are within the radius of 100 km and
within 12 hours of the acquisition of the aircraft profile. Sensitivities of the results to the radius
and time criteria for co-location selection are further investigated in Section 4.2.
(3) For each pair of co-located MOPITT retrieved and in-situ profiles, we apply the MOPITT a
priori profile and averaging kernel to the in-situ profile as in Eq. (1).  Thus, after converting from
profiles of the in-situ and a priori CO concentrations to $\log_{10}$(VMR) profiles ($x_{in-situ}$ and $x_a$), we
calculate
$$x_{transformed} = x_a + A(x_{in-situ} - x_a) \tag{2}$$
so that the $\log_{10}$(VMR)-based transformed in-situ profile ($x_{transformed}$) has the same degree of
smoothing and a priori dependence as the MOPITT retrieved $\log_{10}$(VMR) profile ($x_{rtv}$).
(4) For each in-situ profile, there are likely to be multiple MOPITT retrievals that meet the above
co-location criteria. If fewer than five MOPITT retrievals are co-located with an in-situ profile,
the in-situ profile is not used in the following study and analysis. If an in-situ profile is co-located
with five or more MOPITT retrievals (assume the number to be $N_{retrieval}$), then the following steps
are used in the comparison with MOPITT: (a) the averaging kernel and a prior of each co-located
MOPITT retrieval are applied to the in-situ profile (through equation 2) to obtain $N_{retrieval}$ of
$x_{transformed}$. Note that applying these $N_{retrieval}$ sets of MOPITT a priori profiles and averaging
kernels to the same in-situ profile results in differently transformed in-situ profiles; (b) the $N_{retrieval}$
of $x_{transformed}$ are averaged in $\log_{10}$(VMR) space; and (c) the $N_{retrieval}$ of MOPITT retrievals $x_{rtv}$
are also averaged.

Figure 2 shows an example of profile comparisons (the original aircraft profile, aircraft

profile extended with CAMS reanalysis data and regridded to 35-level grid, $x_{in-situ}$, $x_a$,
$x_{transformed}$, and $x_{rtv}$) in VMR for an aircraft profile sampled on July 22, 2011 during
DISCOVER-AQ in Maryland (MD). Figure 2 also demonstrates what to expect within a MOPITT
retrieval pixel and vertical level. The MOPITT retrievals have a spatial resolution of about 22 km
× 22 km, and each MOPITT retrieval level corresponds to a layer immediately above that level.
The standard deviation of the original aircraft CO observations in each MOPITT layer are also
shown, which is due to horizontal and vertical variability in CO. Taking the 800-hPa layer as an
example, the standard deviation of the original aircraft CO observations in the level is 21.4 ppb,
which is larger than the difference between $x_{transformed}$ and $x_{rtv}$ at that level (12.4 ppb). We also
show the relative scale of the aircraft profile (3 km × 5 km) and a MOPITT pixel (22 km × 22 km)
in Figure 2. We expect the variability of CO within a MOPITT pixel to be even larger than the CO
variability within the scale of 3 km × 5 km. The variability within a satellite pixel and the
representativeness error in the satellite retrieval and aircraft profile comparisons make it
challenging to compare satellite retrievals to aircraft observations. This is one of the major reasons
that MOPITT has yet to be compared with aircraft observations over urban regions with in-situ
observations. The representativeness error has been discussed in previous studies (Fishman et al.,
2011; Follette-Cook et al., 2015; Judd et al., 2019). Follette-Cook et al. (2015) quantified spatial
and temporal variability of column integrated air pollutants, including CO, during DISCOVER-
AQ MD from modeling perspective (using the Weather Research and Forecasting model coupled
with Chemistry – WRF-Chem). They found that during the July 2011 DISCOVER-AQ campaign,
the mean CO difference at the distance of 20-24 km is ~30 ppb (derived from the aircraft
observations) and ~40 ppb (derived from co-located WRF-Chem output), based on structure
function analyses. In this study, we demonstrate this challenge with an example in Figure 2. We
also show a sensitivity analysis in Section 4 to provide perspectives on how the spatial and
temporal representativeness may change the results. Further quantification of the variability within
MOPITT pixels would be very challenging (partially due to limited coverage of the observational
data), and we will elaborate more on this issue in Section 5.

**3. MOPITT comparisons with aircraft profiles over urban and non-urban regions**
In this section, the results for MOPITT comparisons with aircraft profiles are provided for
only daytime retrievals (i.e., solar zenith angle < 80° in the retrieval), because (1) MOPITT
retrievals generally contain more CO profile information in daytime, which is reflected in AKs
and Degrees of Freedom for Signal (DFS) in Figure 3, and (2) most aircraft profiles are sampled
during daytime. In Section 4.3, we discuss the sensitivity to the inclusion of MOPITT nighttime
retrievals in MOPITT comparisons with aircraft profiles. In addition, many aircraft profiles,
especially those from DISCOVER-AQ, lack observations above 600 hPa. Even though we
extended the aircraft profiles vertically with reanalysis data (as discussed in Section 2.3), this still
prevents the use of these profiles for validating MOPITT retrievals at upper levels against in-situ
observations. In this paper, we only focus on comparing MOPITT retrievals below the altitude of
600 hPa to aircraft profiles. Nevertheless, since the CO retrievals below 600 hPa are still weakly
impacted by CO fields in the upper levels (as shown by the AKs in Figure 3), in Section 4.1 we
perform sensitivity tests on how augmenting the aircraft profiles with reanalysis fields affects the
comparison results.
**3.1 Overall statistics**
The overall comparison results are presented in Table 2. Following Deeter et al. (2017),
retrieval biases and standard deviation (SD) are calculated based on mean $x_{rtv}$ and $x_{transformed}$
for each in-situ profile, and converted from $\log_{10}$(VMR) to percent. The correlation coefficient (r)
is quantified based on $(x_{rtv} - x_a)$ and the corresponding $(x_{transformed} - x_a)$ to avoid
correlations which mainly result from the variability of the a priori. $x_{rtv}$, $x_{transformed}$, and $x_a$ are
in log$_{10}$(VMR) space in order to apply the AKs, which are computed for $x_{rtv}$ in log$_{10}$(VMR). These
comparisons for MOPITT Version 8 TIR-only (V8T) and Version 8 TIR-NIR (V8J) are shown in
Figures 4 (for all profiles) and 5 (for urban profiles). Overall biases for V8J products (averaged
over all campaigns in Table 1) vary from -0.7% to 0.0%, which are lower than biases for V8T
(from 2.0% to 3.5%). Overall biases for V8J products are also smaller than biases for V7J (from -
0.5% to -5.4%). For V8J and V7J, biases over urban regions vary from -0.8% to -2% and from -
1.4% to -8.9%, respectively, which are generally larger than biases over non-urban regions (-
0.3%~1.1% and -3.3%~0.1%). Correlation coefficients over non-urban regions are higher than
those over urban regions for all six products (V7T, V8T, V7N, V8N, V7J, V8J) at all three levels
in Table 2 (the surface layer, the 800-hPa layer, and the 600-hPa layer). We also notice that for
TIR-NIR and TIR-only products, V8 have higher correlation coefficients with in-situ
measurements than V7 over non-urban regions, whereas over urban regions, V8 products have
lower correlation coefficients than V7 (except for the 600-hPa layer). Overall, MOPITT products
(especially V8J) perform reasonably well over both urban and non-urban regions. Performance
over non-urban regions is better than that over urban regions in terms of higher correlation
coefficients and smaller biases for V8J and V7J.
**3.2 Discussions on individual campaigns**
We also evaluate MOPITT V8J retrievals during individual field campaigns with results in
Figure 6. The corresponding results for MOPITT V8T are summarized in Figure S2. The patterns
of biases are very similar for MOPITT V8J and V8T. Thus, in this sub-section, we focus on V8J
unless stated otherwise. Overall, except comparisons with A-FORCE and ARIAs, biases over
urban regions and non-urban regions do not have a significant difference. Neither do biases
determined for campaigns over the US and East Asia differ significantly, either.
When compared to DISCOVER-AQ CA, MOPITT CO values are generally higher than
in-situ profiles at the 600-hPa layer (i.e., the 100 hPa uniform layer immediately above 600 hPa)
but not at the surface layer (i.e., the uniform layer immediately above the surface). This is likely
related to the fact that the DISCOVER-AQ CA aircraft profiles are mostly below 600 hPa, and
hence CO values of these in-situ profiles at 600 hPa and above are filled with CAMS reanalysis
data. In addition, DISCOVER-AQ CA was conducted in the winter when boundary layer height is
at lower altitudes, which could also explain the difference, in particular since most of the other
campaigns are during times with greater vertical mixing. The lack of aircraft observations at 600
hPa and above also has a smaller impact on the biases at the 800-hPa layer through applying AK
(see Figure 3).
During the A-FORCE campaign, only 2 in-situ profiles out of 45 were sampled over urban
regions. The locations of the two profiles are close to each other and they are both sampled on/near
the coast of South Korea (Figure 1). MOPITT has large negative biases (-30%~-40%) when
compared to these two profiles. The averaged $x_{in-situ}$, $x_a$, $x_{transformed}$, and $x_{rtv}$ over non-urban
regions during A-FORCE and the $x_{in-situ}$, $x_a$, $x_{transformed}$, and $x_{rtv}$ of the two profiles over
urban regions are shown in Figure S3. Compared to the averaged $x_{in-situ}$ over non-urban regions,
the $x_{in-situ}$ for the two profiles over the urban regions have large enhancements near the surface
and between 600~800 hPa. Even though the $x_a$ and $x_{rtv}$ for the two profiles have higher CO
concentrations (~400 ppb at the surface layer) than the averaged $x_a$ and $x_{rtv}$ (~200 ppb at the
surface layer), they are still lower than the $x_{transformed}$.
As for KORUS-AQ, MOPITT also has a negative bias (though smaller) when compared to
the profiles over urban regions. Most of these KORUS-AQ profiles were located near the two
profiles from A-FORCE but farther from the coast. The negative bias is not seen over non-urban
regions during KORUS-AQ at the surface layer.
When compared to the in-situ profiles from ARIAs, MOPITT has a large positive bias,
especially over urban regions (20%~30%). During ARIAs, in-situ profiles over urban regions have
lower CO values (~200 ppb at the surface layer) than those in-situ profiles over non-urban regions
(~ 400 ppb at the surface layer; Figure S4). We note there are only a small number of in-situ
profiles over urban regions in East Asia used in this study, compared to what is provided by
DISCOVER-AQ in the US. The large negative biases against A-FORCE and large positive biases
against ARIAs point to the need for more in-situ observations over East Asia.
**3.3 MOPITT comparisons with aircraft profiles at high CO concentrations**
Urban regions are often associated with high CO concentrations. But this is not always the
case (e.g., Figure S4). Here we separate the in-situ profiles at the surface layer, the 800-hPa layer,
and the 600-hPa layer into lower 50% CO values and higher 50% CO values based on CO values
at each level to demonstrate the impact of CO concentrations on the MOPITT product validation
(Figure 7). For V8J, MOPITT has smaller biases at higher 50% CO concentrations for all three
levels, whereas for V8T, MOPITT has larger biases at the surface layer and the 600-hPa layer at
higher 50% CO concentrations. For the higher 50% of measured mixing ratios both V8J and V8T
have larger SDs and lower correlation coefficients at the surface layer, the 800-hPa layer, and the
600-hPa layer, suggesting that the agreement between MOPITT and the in-situ profiles at higher
CO concentrations is not as good as that at lower CO concentrations. In contrast, Deeter et al.
(2016) found that the retrieval biases do not visibly increase at the upper range of CO
concentrations when compared to aircraft measurements over the Amazon basin. The vertical error
bars in Figure 7 (caused by the multiple co-located MOPITT profiles with one in-situ profile)
represent the variability (standard deviation) of the MOPITT data used to calculate each of the
plotted mean values. For an in-situ profile, the variability of the MOPITT data located within its
radius of 100 km and within 12 hours is larger when the in-situ profile has higher CO values,
indicated by larger error bars at higher 50% CO concentrations. At higher 50% CO concentrations,
the averaged retrieval uncertainties for the 600-hPa, 800-hPa, and surface layers, are 28%, 28%,
and 29%, respectively. This is smaller than the averaged retrieval uncertainties at lower 50% CO
concentrations (28%, 29%, and 30% for the 600-hPa, 800-hPa, and surface layers, respectively).
We therefore conclude that the larger apparent biases at high CO concentrations are related to
greater CO variability and representativeness error of the in-situ profile within the co-location
radius used for analyzing the MOPITT data, rather than indicating larger retrieval uncertainties.
Theoretically, MOPITT retrievals perform better with higher CO concentrations. The larger biases
at high CO concentrations in Figure 7 implies that the relatively greater CO variability may
overcome the impact of high CO concentrations. Addressing representativeness error and spatial
variability in the comparisons between satellite and in-situ profiles is challenging, and will be
discussed further in Section 5.

We will discuss the sensitivity of radius and time difference for the selection of co-located

data in Section 4. The difference in the variability at different CO concentrations was not found in
Deeter et al. (2016). It could be partially due to the fact that the aircraft profiles over the Amazon
basin used in Deeter et al. (2016) were sampled under more geographically homogeneous
conditions, whereas the profiles used in this study are from different campaigns, and high CO
concentrations over and near urban regions might be associated with more complex and
inhomogeneous conditions.

### 4. Sensitivities to assumptions made for aircraft-satellite comparisons

In Section 3, we compared profiles over urban and non-urban regions separately to
MOPITT V8T, V8N, V8J, V7T, V7N, and V7J. In this section, we compare only the MOPITT
V8J product to all the in-situ profiles (both over urban and non-urban regions) described in Table
1 to test the sensitivity of results to the assumptions made during the comparisons.

### 4.1 Sensitivity to the in-situ profile extension

As discussed in Section 2.3, the in-situ profiles must be vertically extrapolated or extended
to compare with MOPITT products due to aircraft altitude limits.  Thus, model or reanalysis data
must be merged with the in-situ data to generate a complete CO profile for comparisons with
MOPITT satellite retrievals. The use of model or reanalysis data may introduce uncertainties in
the comparison results as they are not measured directly. The parameter $P_{interp}$ controls the impact
of the model-based profile extension on the shape and value of in-situ profiles (see Figure S5).
Here we test the sensitivity of validation results to various $P_{interp}$ values (100 hPa, 200 hPa, 300
hPa, 400 hPa, 500 hPa) to demonstrate the potential impact of the profile extension. Note that the
model-based profile extension and the value of $P_{interp}$ impacts the validation results through
changing the augmented observational profile, which is different from the other sensitivity tests in
this study that change the selection of MOPITT data. The agreements between the values of
MOPITT and in-situ profiles at the surface layer are insensitive to the selection of $P_{interp}$ (Figure
8). The overall agreements between the values of MOPITT and in-situ profiles at the 800-hPa layer
are also not sensitive to $P_{interp}$, except for the results against DISCOVER-AQ CA which have
slightly larger biases when $P_{interp}$ is 200 hPa or 100 hPa since the DISCOVER-AQ CA aircraft
profiles at 600 hPa and above are mostly extended using reanalysis data. Therefore, the
comparisons with DISCOVER-AQ CA are more likely to be affected by $P_{interp}$ compared to other
campaigns which typically obtained higher maximum aircraft altitudes. At the 600-hPa layer, the
agreements between the values of MOPITT and in-situ profiles are affected more by $P_{interp}$
compared to the those at the surface layer and the 800-hPa layer for comparisons with all the
campaigns. The overall validation results using 100 hPa as $P_{interp}$ have larger biases than using
other values of $P_{interp}$. At 400-hPa layer and 200-hPa layer, the comparisons are even more sensitive
to $P_{interp}$ for all the campaigns (Figure S6). The CAMS 3-hourly reanalysis data are constrained by
observations, but its usage may still introduce the uncertainties in the validation results especially
at upper pressure levels (e.g., 200 hPa and 400 hPa). Previous MOPITT evaluation results may be
subject to larger uncertainties by using CAM-chem monthly CO fields that are not constrained by
observations (e.g., Deeter et al., 2012, 2016).

**4.2 Sensitivity to the radius and allowed maximum time difference as criteria for co-location**

The criteria for co-location in this study (within a radius of 100 km and within 12 hours of
the acquisition of the aircraft profile) generally follow previous MOPITT validation studies (e.g.,
Deeter et al., 2016, 2019) and are chosen empirically. They are selected based on a trade-off
between uncertainties generated from CO spatial and/or temporal variability, and the number of
included MOPITT retrievals that impacts the statistical robustness. Here we test the sensitivity of
the results to the two criteria for co-location. The boxplot of biases calculated with different radii
(200 km, 100 km, 50 km, and 25 km) at the surface layer, the 800-hPa layer, and the 600-hPa layer
are shown in Figure 9. Overall, the biases calculated with radius of 200 km, 100 km and 50 km are
similar, whereas the biases calculated with the radius of 25 km are different from others. The
comparisons of MOPITT to in-situ profiles results using the radius of 25 km generally have larger
biases and SD, due to including fewer MOPITT retrievals. In some cases, there are no matched
MOPITT retrievals within the radius of 25 km of the aircraft profile (e.g., DISCOVER-AQ CA
and ARIAs). In addition, representativeness errors would be expected to go up if there are only a
few retrievals over a more polluted and perhaps heterogeneous area. We note that the usage of the
largest radius (200 km) in this paper does not appear to degrade the overall results, even though
representativeness errors generated from CO spatial and/or temporal variability are expected to
increase. However, the use of the smallest radius (25 km) degrades the overall results by reducing
the number of included MOPITT retrievals.
The boxplot of biases calculated with four sets of allowed maximum time difference (12
hours, 6 hours, 3 hours, and 1 hours) are shown in Figure 10. The overall results are not sensitive
to the selection of allowed maximum time difference. One exception is the comparisons to the
SEAC[4]RS campaign at the 600-hPa layer, due to a smaller number of MOPITT retrievals in the
shorter time window. We note that when comparing to the ARIAs campaign, using 1h as the
allowed maximum time difference decreases the biases at the surface layer, the 800-hPa layer, and
the 600-hPa layer, compared to the cases using longer allowed maximum time difference (i.e., 3h,
6h, and 12h). This implies that the temporal variability is relatively large in the region. And the
improvement observed for ARIAs for the shortest time also points to the possibility that short term
emission sources might be responsible for the large biases there. On the other hand, when the
allowed maximum time difference equals 1 hour, there are only 6 aircraft profiles that have
matched MOPITT retrievals.

**4.3 Sensitivity to the inclusion of MOPITT nighttime retrievals**


Previous MOPITT validation studies have only included MOPITT daytime observations.
Over land, MOPITT retrievals for daytime and nighttime overpasses are characterized by
significantly different averaging kernels (Figure 3), and may be subject to different types of
retrieval error (Deeter et al., 2007). CO has a long enough lifetime (approximately a month;
Gamnitzer et al., 2006) in the free troposphere that nighttime observations could be potentially
comparable, in general, to the daytime flights for remote sites. However, for urban regions where
the spatiotemporal variability of the emissions and evolution of the planetary boundary layer drives
large changes in the measured CO, comparisons of MOPITT nighttime observations to aircraft
profiles sampled during daytime may introduce representative uncertainties, especially for areas
that are subject to strong nocturnal inversions and the surface CO can be enhanced. It is difficult
to disentangle the effects of the MOPITT daytime/nighttime performance and the uncertainty from
the temporal representativeness, based on the comparison of the MOPITT daytime/nighttime
retrievals with daytime aircraft profiles. Therefore, we only include the results in Figure S7 and
briefly describe the results here without drawing any further conclusions. Overall, MOPITT
nighttime retrievals have larger biases than daytime retrievals, which could be expected since most
of the aircraft profiles are sampled during daytime. Flight campaigns with nighttime observations
are needed to validate MOPITT nighttime retrievals.

**4.4 Sensitivity to the signal-to-noise ratio (SNR) filters**


The MOPITT Level 3 data are generated from Level 2 data, and are available as gridded
($1° \times 1°$) daily-mean and monthly-mean files. Pixel filtering and signal-to-noise ratio (SNR)
thresholds for Channel 5 and 6 Average radiances are used when averaging Level 2 data into Level
3 data, and this increases overall mean DFS values (details can be found in the MOPITT Version
8 Product User's Guide, 2018). Taking MOPITT V8J daytime product as an example, the Level 3
data product excludes all observations from Pixel 3 (one of the four elements of MOPITT's linear
detector array that has highly variable Channel 7 SNR values), or observations where both the
Channel 5 Average radiances SNR < 1000 and the Channel 6 Average radiances SNR < 400. In
Figure 11, we test the impact of applying the aforementioned SNR filters on the agreement between
MOPITT and in-situ profiles. Note that we are not suggesting the comparisons between MOPITT
Level 3 product and aircraft measurements. Because the MOPITT Level 3 product is gridded data
and represent the average value in a 1°×1° grid. Comparing the grid average value to an aircraft
profile within it may be subject to large representativeness errors. Here we only show the
sensitivity of agreement between MOPITT Level 2 data and aircraft profiles to the application
SNR filter. We find that applying the SNR filters does not significantly change the overall
agreement between MOPITT retrievals and the in-situ profiles used in this study. This is mostly
because applying the SNR filters reduces the number of MOPITT retrievals included in the
comparisons. This effect is particularly important if there are not many MOPITT retrievals to begin
with (such as our comparisons with in-situ profiles in this study). Even though applying SNR filter
when generating Level 3 data does not significantly change the agreement with the in-situ profiles
used in this study, by excluding low-SNR observations from the Level 3 cell-averaged values
raises overall mean DFS values (MOPITT Version 8 Product User's Guide, 2018). In addition, the
Level 3 product typically are less affected by random retrieval errors (e.g., due to instrument noise
or geophysical noise).

**5. Discussion and conclusions**

MOPITT products are widely used for monitoring and analyzing CO over urban regions.

However, systematic validation against observations over urban regions has been lacking. In this
study, we compared MOPITT products over urban regions to aircraft measurements from
DISCOVER-AQ, SEAC[4]RS, ARIAs, A-FORCE, and KORUS-AQ campaigns. The DISCOVER-
AQ campaign was designed primarily with satellite validation in mind, and the campaign over MD,
CA, TX, and CO together contributes 64.8% (232 out of 358) of the aircraft profiles and 91.0%
(121 out of 133) of the aircraft profiles over the urban regions in this study (Table 1). Therefore,
the DISCOVER-AQ campaign largely contributes to the results and the statistics in this study. We
found that MOPITT mean biases are well within the 10% required accuracy (Drummond and Mand,
1996) for both urban and non-urban regions (mean biases for V8J and V8T vary from -0.7% to
0.0%, and from 2.0% to 3.5% for different levels). The performance over non-urban regions is
better than that over urban regions in terms of correlation coefficients for the 6 products in Table
2, and biases of V8J and V7J. However, the in-situ profiles over East Asia used in this study are
limited, especially over urban regions (only 11 profiles). The large biases against aircraft profiles
from the A-FORCE and ARIAs campaigns point to the need for more in-situ observations over
East Asia. We also studied the impact of CO concentrations on the agreement between MOPITT
products and in-situ profiles by dividing the aircraft profiles of CO into two groups of high CO
(upper 50%) and low CO (lower 50%). We found that MOPITT retrievals at high CO
concentrations have higher biases and lower correlations compared with low CO concentrations,
although CO variability may tend to exaggerate retrieval biases in heavily-polluted scenes. The
statistics are often very similar between different versions and products over urban and non-urban
regions, and in general, MOPITT agrees reasonably well with the in-situ profiles in both cases.
There is not, therefore, any reason to recommend the continued use of MOPITT versions earlier
than V8 based on urban or non-urban region considerations. In general, MOPITT V8 is
recommended (Deeter et al., 2019) as it uses a new parameterized radiance bias correction method
to minimize retrieval biases, and has updated spectroscopic data for water vapor and nitrogen.

In addition, the assumptions and data filters made during aircraft-satellite comparisons may

impact the validation results. We tested the sensitivities of the results to assumptions and data
filters, including the model-based extension to the in-situ profile, radius and allowed maximum
time difference as criteria for the selection of co-located data, the inclusion of nighttime MOPITT
data, and the SNR filters. The agreements between the values of MOPITT and in-situ profiles at
the surface layer are insensitive to the model-based profile extension, whereas the results at upper
levels (e.g., 400 hPa and 200 hPa) are more sensitive to the profile extension, as there are very
limited aircraft observations. The results are insensitive to the allowed maximum time difference
as a co-location criteria, and are generally insensitive to the radius for co-location except for the
case with a radius of 25 km, where only a small number of MOPITT retrievals are included in the
comparisons. Overall, daytime MOPITT products overall have smaller biases than nighttime
MOPITT products. However, conclusions regarding the performance of MOPITT daytime and
nighttime retrievals cannot be drawn due to the fact that most of the aircraft profiles are sampled
during daytime. As we mentioned earlier, MOPITT daytime and nighttime retrievals may be
subject to different retrieval errors. In addition, previous studies suggest pollutants themselves may
have different characteristics during daytime and nighttime (e.g., Yan et al., 2018). Therefore,
validation of MOPITT nighttime retrievals, with a sufficient number of nighttime airborne profiles,
is needed in order to study nighttime CO characteristics and trends. Applying SNR filters does not
necessarily change the overall agreement between MOPITT retrievals and in-situ profiles used in
this study significantly, and this may be partially caused by the smaller number of MOPITT
retrievals in the comparisons after the SNR filters. We note that comparisons to ARIAs are
exceptional in a few sensitivity tests due to rather a limited number of aircraft measurements.
Given the large biases against aircraft profiles from the ARIAs campaign, more in-situ
observations over East Asia especially China are needed in order to validate MOPITT products in
the region.
Validation and evaluation of satellite retrievals with aircraft observations are very
challenging, and assumptions have to be made for the comparisons. As discussed in Section 2, the
CO spatial variability within MOPITT retrieval pixels and the representativeness error of aircraft
profiles when compared to MOPITT retrievals may introduce uncertainties in the validation
results. This issue is difficult to address and quantify due to the limited spatial coverage of dense
aircraft observations. One possible way is to study $NO_2$ data retrieved from the Geostationary
Trace Gas and Aerosol Sensor Optimization (GeoTASO) at very high resolution ($250\,\mathrm{m} \times 250\,\mathrm{m}$),
to provide an upper estimate on CO variability. Besides, the variability of Tropospheric Monitoring
Instrument (TROPOMI) CO retrievals (resolution: 7 km×7 km; Landgraf et al., 2016) might also
provide information on MOPITT sub-pixel variability. Further research on trace gas spatial
variability within satellite retrieval pixels, and quantification of the representativeness error
incurred by comparing individual aircraft profiles to satellite products is needed, and will be the
subject of a follow-up study.

**Data availability**
MOPITT products are available at https://www2.acom.ucar.edu/mopitt (Last access date:
January 14[th], 2020). MOPITT Version 8 Product User's Guide is available online at
https://www2.acom.ucar.edu/sites/default/files/mopitt/v8_users_guide_201812.pdf (Last access

date: January 15[th], 2020). DISCOVER-AQ data can be accessed at https://www-air.larc.nasa.gov/missions/discover-aq/discover-aq.html (Last access date: January 14[th], 2020). SEAC$_4$RS data can be accessed at https://www-air.larc.nasa.gov/missions/seac4rs/ (Last access date: January 14[th], 2020). KORUS-AQ and ARIAs data can be accessed at https://www-air.larc.nasa.gov/missions/korus-aq/index.html (Last access date: January 14[th], 2020). A-FORCE data are available upon request (Yutaka Kondo: kondo.yutaka@nipr.ac.jp). MODIS Land Cover Type Global product (MCD12C1 v006) is available at https://earthdata.nasa.gov/ (Last access date: January 14[th], 2020).

**Author contribution**

WT, HMW, and MND designed the study. WT analyzed the data with help from MND, SMA, and LKE. GSD provided CO measurements during DISCOVER-AQ SEAC$_4$RS, and KORUS-AQ. RRD, XR, and HH provided CO measurements during ARIAs. YK provided CO measurements during A-FORCE. HMW, MND, DPE, LKE, BG, RRB, and XR offered valuable discussions and comments in improving the study. WT prepared the manuscript with improvements from all the other coauthors.

**Acknowledgements**

The National Center for Atmospheric Research (NCAR) is sponsored by the National Science Foundation (NSF). W. Tang is supported by a NCAR Advanced Study Program Postdoctoral Fellowship. The NCAR MOPITT project is supported by the National Aeronautics and Space Administration (NASA) Earth Observing System (EOS) Program. The authors thank the DISCOVER-AQ, SEAC$_4$RS, ARIAs, A-FORCE, and KORUS-AQ Science Teams for the valuable in-situ observations. We thank Drs. Naga Oshima and Makoto Koike for the A-FORCE data. ARIAs was supported by NSF (grant # 1558259) and National Institute of Standards and Technology (NIST, grant #70NANB14H332). The authors thank Dr. Frank Flocke for helpful comments on the manuscript. Wenfu Tang thanks Dr. Cenlin He for helpful discussions.

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

**Table 1.** In-situ datasets of CO used for MOPITT products validation in this study.

| | Period | Region | Number of profiles | Number of profiles over urban | Technique | Uncertainty | Reference |
|---|---|---|---|---|---|---|---|
| DISCOVER-AQ MD | Jul, 2011 | Baltimore-Washington, D.C., US | 80 | 36 | NASA DACOM | | |
| DISCOVER-AQ CA | Jan-Feb, 2013 | California, US | 35 | 12 | NASA DACOM | Precision < 1% or 0.1 ppbv; Accuracy 2% | https://www-air.larc.nasa.gov/ missions/discover-aq/ |
| DISCOVER-AQ TX | Sep, 2013 | Texas, US | 61 | 37 | NASA DACOM | | |
| DISCOVER-AQ CO | Jul-Aug, 2014 | Colorado, US | 56 | 36 | NASA DACOM | | |
| SEAC$^4$RS | Aug-Sep, 2013 | US | 15 | 1 | NASA DACOM | Precision < 1% or 0.1 ppbv; Accuracy 2% | Toon et al. (2016) |
| A-FORCE | Mar-Apr, 2009; Feb-Mar, 2013; Jun-Jul, 2013 | Japan, South Korea, Pacific Ocean | 45 | 2 | AL5002, Aero-Laser GmbH | Precision ~0.5%; Accuracy 2% | Oshima et al. (2012); Kondo et al. (2016) |
| KORUS-AQ | May-Jun, 2016 | South Korea | 47 | 6 | NASA DACOM | Precision < 1% or 0.1 ppbv; Accuracy 2% | Al-Saadi et al. (2015) |
| ARIAs | May-Jun, 2016 | Hebei, East China | 19 | 3 | Picarro G2401-m | Precision of ±4 ppbv | Wang et al. (2018) |

The CO scale used for SEAC$^4$RS, and DISCOVER-AQ MD, TX, and CA is WMO-CO-X2004, while the CO scale
used for ARIAs and KORUS-AQ  is WMO-CO-X2014A.
**Table 2.** Summarized validation results for V7 and V8 TIR-only (V7T and V8T), NIR-only (V7N
and V8N) and TIR-NIR (V7J and V8J) products based on in-situ profiles from DISCOVER-AQ,
SEAC[4]RS, A-FORCE, KORUS-AQ, and ARIAs.

| | | Surface layer | | | 800-hPa layer | | | 600-hPa layer | | |
|---|---|---|---|---|---|---|---|---|---|---|
| | | All | Urban | Non-urban | All | Urban | Non-urban | All | Urban | Non-urban |
| V7T | Bias (%) | 0.1 | -1.7 | 1.1 | 0.8 | -0.6 | 1.7 | 4.0 | 3.9 | 4.0 |
| | SD (%) | 9.5 | 8.6 | 9.8 | 11.0 | 9.0 | 11.9 | 11.4 | 9.0 | 12.7 |
| | r | 0.71 | 0.67 | 0.72 | 0.66 | 0.65 | 0.66 | 0.63 | 0.58 | 0.64 |
| V8T | Bias (%) | 2.0 | 0.9 | 2.7 | 2.2 | 1.4 | 2.7 | 3.5 | 3.5 | 3.5 |
| | SD (%) | 9.3 | 9.6 | 9.0 | 10.7 | 9.7 | 11.2 | 11.7 | 10.0 | 12.6 |
| | r | 0.70 | 0.58 | 0.75 | 0.66 | 0.58 | 0.69 | 0.63 | 0.54 | 0.66 |
| V7N | Bias (%) | -2.0 | -2.8 | -1.5 | -1.6 | -2.1 | -1.1 | -1.6 | -1.9 | -1.3 |
| | SD (%) | 6.7 | 6.4 | 6.9 | 5.7 | 5.2 | 6.0 | 4.3 | 4.2 | 4.4 |
| | r | 0.62 | 0.54 | 0.67 | 0.56 | 0.45 | 0.61 | 0.61 | 0.48 | 0.68 |
| V8N | Bias (%) | 1.4 | 0.4 | 2.2 | 1.6 | 0.9 | 2.1 | 1.2 | 0.8 | 1.5 |
| | SD (%) | 6.9 | 6.7 | 6.9 | 6.0 | 5.8 | 6.1 | 4.6 | 4.7 | 4.5 |
| | r | 0.60 | 0.52 | 0.67 | 0.54 | 0.40 | 0.62 | 0.59 | 0.42 | 0.68 |
| V7J | Bias (%) | -5.4 | -8.9 | -3.3 | -3.9 | -6.5 | -2.4 | -0.5 | -1.4 | 0.1 |
| | SD (%) | 13.5 | 12.1 | 13.9 | 14.2 | 12.4 | 15.0 | 13.6 | 11.0 | 14.8 |
| | r | 0.68 | 0.63 | 0.70 | 0.64 | 0.58 | 0.66 | 0.60 | 0.52 | 0.62 |
| V8J | Bias (%) | 0.0 | -2.0 | 1.1 | -0.7 | -1.6 | -0.1 | -0.5 | -0.8 | -0.3 |
| | SD (%) | 12.7 | 13.7 | 12.0 | 12.9 | 12.5 | 13.1 | 12.8 | 10.9 | 13.8 |
| | r | 0.69 | 0.53 | 0.76 | 0.69 | 0.57 | 0.73 | 0.65 | 0.53 | 0.67 |



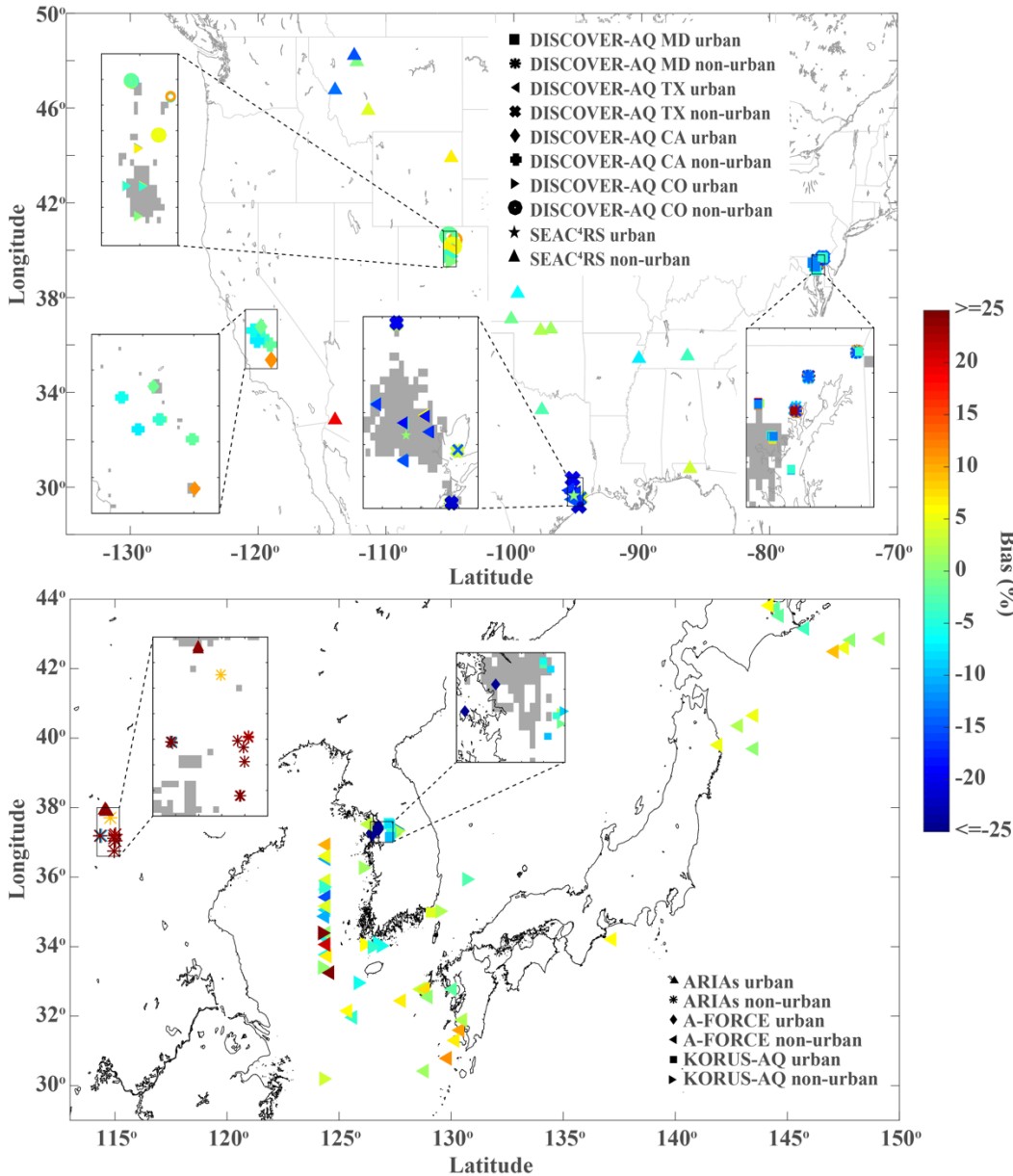

**Figure 1**. Spatial distributions of aircraft profiles from the DISCOVER-AQ, SEAC[4]RS, ARIAs,
A-FORCE, and KORUS-AQ campaigns. Urban and built-up land cover (from MCD12C1 v006)
are shown by gray shade in the boxes. Biases of MOPITT V8J comparing to the aircraft profile at
the surface layer are shown by the color of the profile.

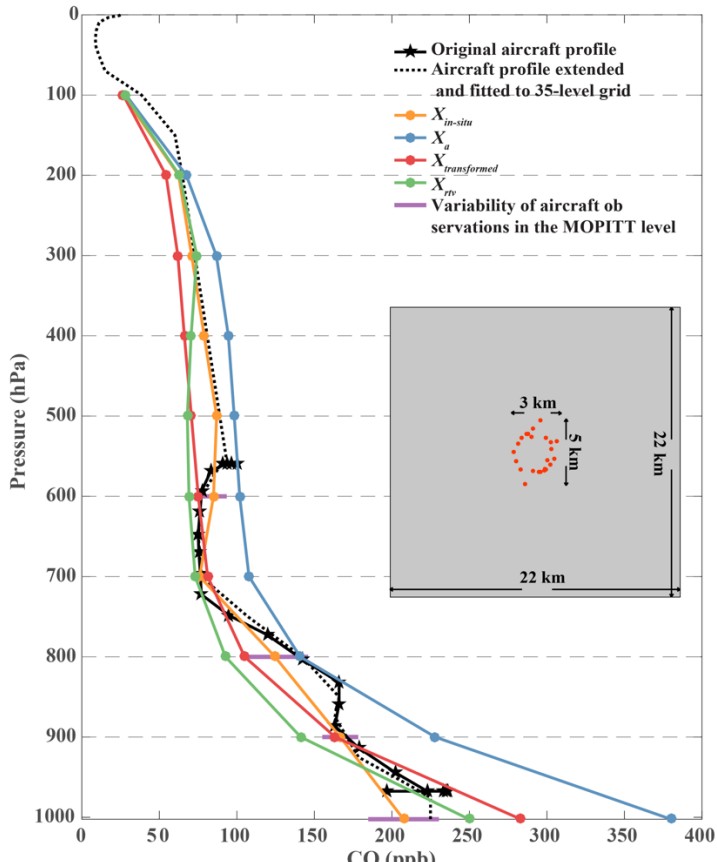

**Figure 2**. Example of profile comparisons for an aircraft profile sampled on July 22, 2011 during
DISCOVER-AQ MD. The black solid line represents the original aircraft profile and the stars
represent the original aircraft observations, the black dotted line is the aircraft profile extended
with CAMS reanalysis data, and regridded to 35-level grid. The in-situ profile regridded at 10-
level grid ($x_{in-situ}$), the MOPITT a priori profile ($x_a$), the in-situ profile transformed with the
MOPITT a priori and AK ($x_{transformed}$), and the MOPITT retrieved profile ($x_{rtv}$) are shown in
colored lines with dots. The purple bars centered at the $x_{in-situ}$ at each MOPITT retrieval level
show the standard deviations of the original aircraft observations in the MOPITT layer. Note that
each MOPITT retrieval level corresponds to a uniform layer immediately above that level.
Superimposed gray box shows the horizontal scale of the profile (each aircraft observation is
represented by a red dot) and a MOPITT pixel (gray box).

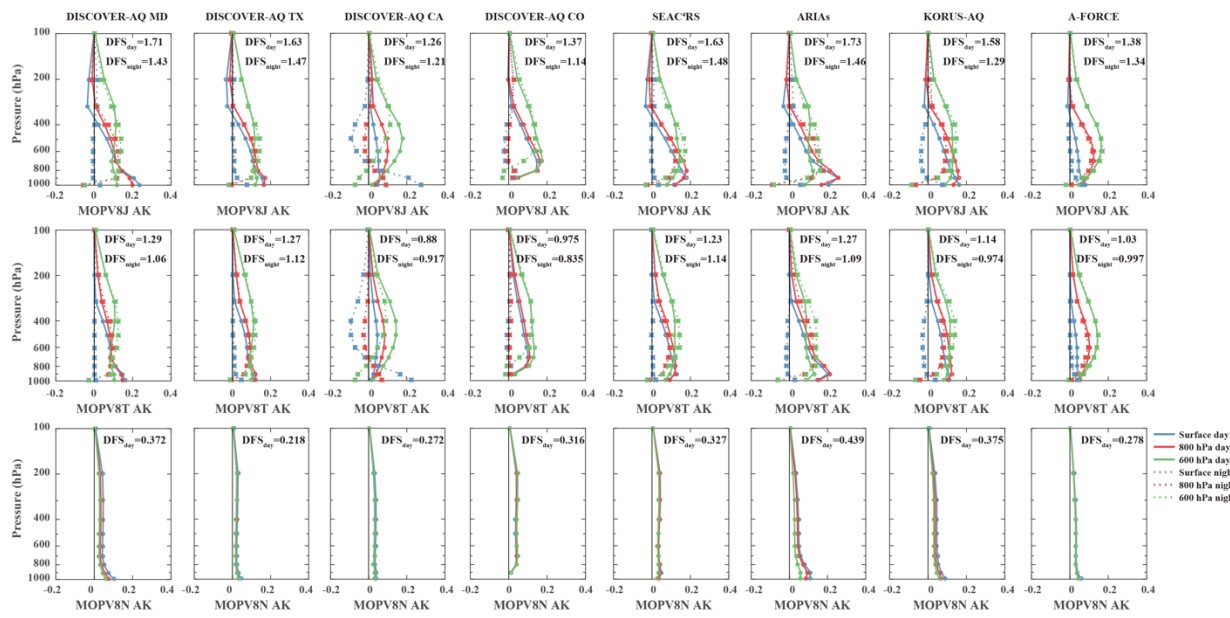

**Figure 3**. Mean retrieval averaging kernels for the MOPITT V8J, V8T, and V8N for the
corresponding in-situ profiles from the DISCOVER-AQ, SEAC[4]RS, ARIAs, KORUS-AQ, and A-
FORCE at daytime (solid lines) and nighttime (dashed lines).

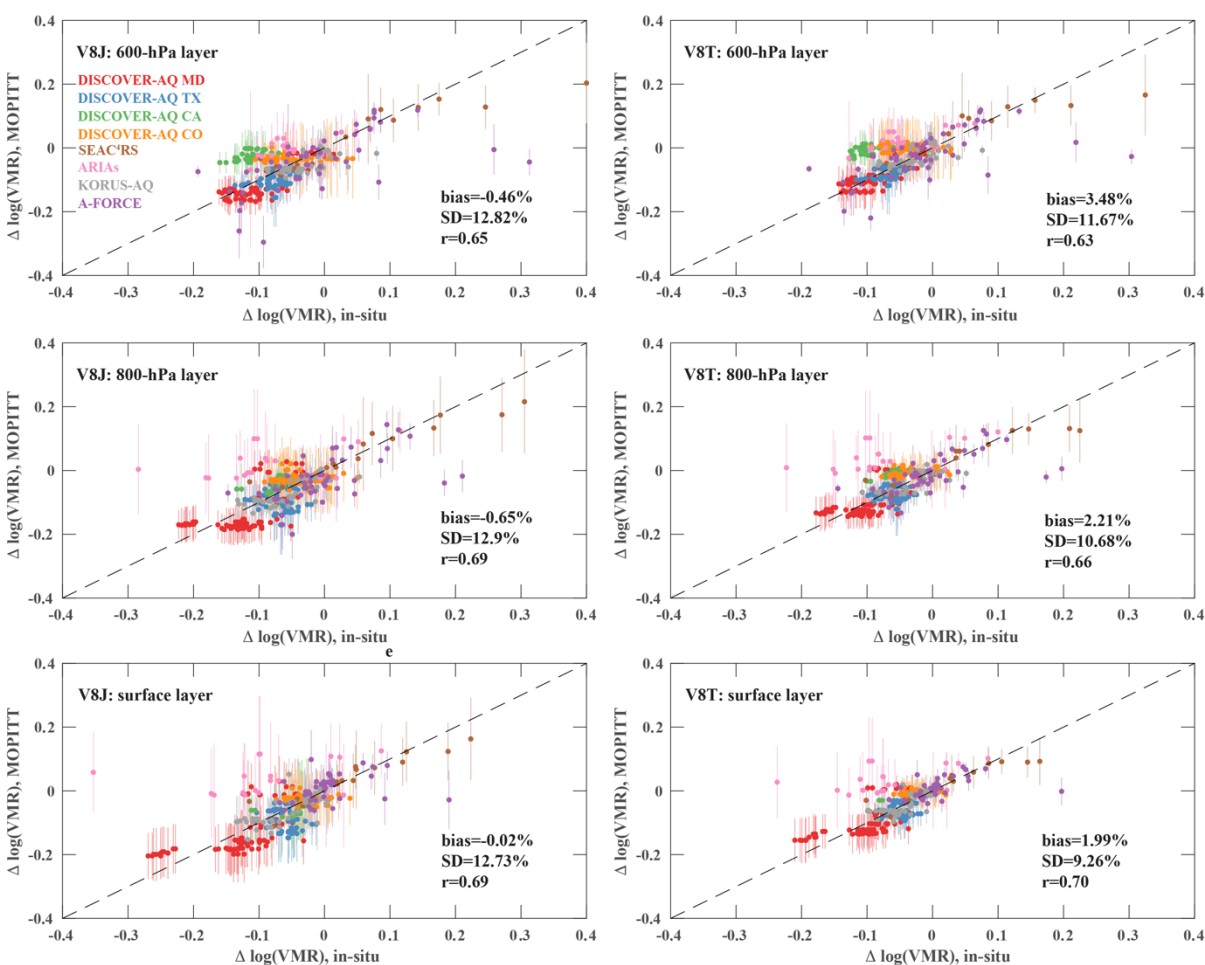

**Figure 4**. MOPITT V8J and V8T validation results over both urban and non-urban regions at 600-
hPa layer, 800-hPa layer, and the surface layer in terms of $\Delta\log_{10}(VMR)$. $\Delta\log_{10}(VMR)$ is defined
as $x_{rtv}$ - $x_a$ for MOPITT profiles and $x_{transformed}$ - $x_a$ for the in-situ profiles. The use of
$\Delta\log_{10}(VMR)$ allows us to remove the impact of the a priori in the comparisons. The variability
of the MOPITT data used to calculate each of the plotted mean values are represented by the
vertical error bars. The dashed lines are one-to-one ratio lines.

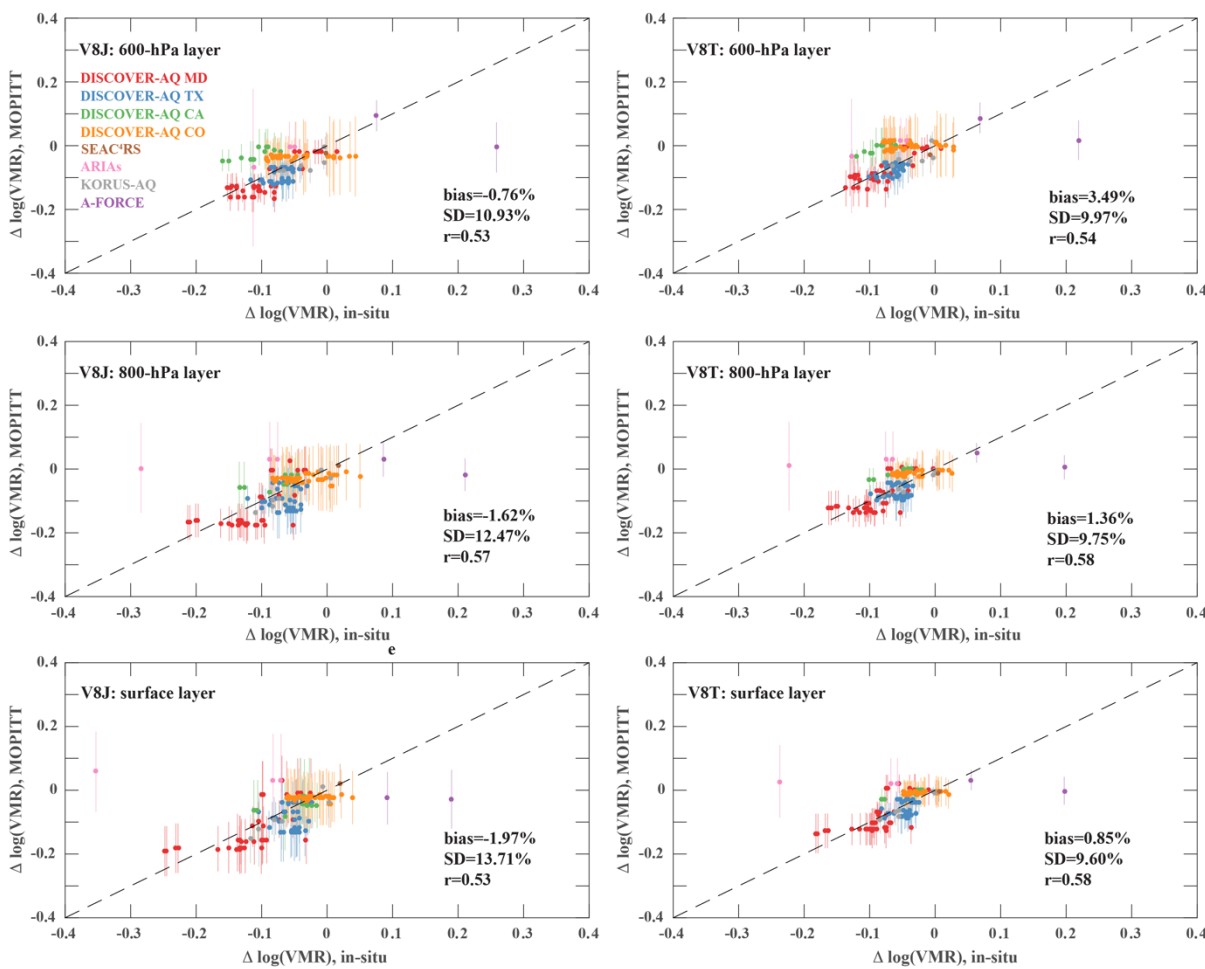

**Figure 5**. MOPITT V8J and V8T validation results against aircraft profiles over urban regions at
the 600-hPa layer, the 800-hPa layer, and the surface layer in terms of Δlog (VMR). The dashed
lines are one-to-one ratio lines. See the caption of Figure 4.


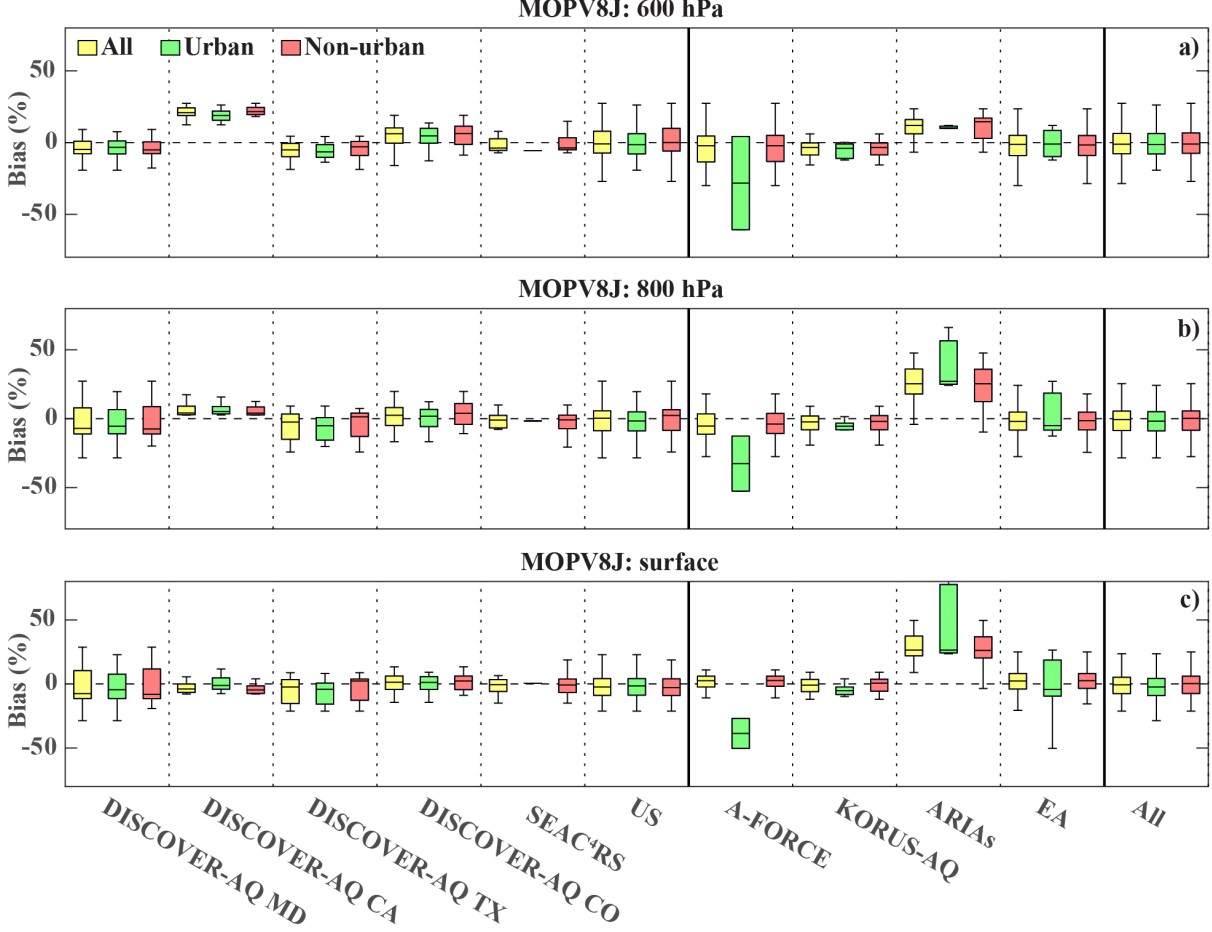

**Figure 6**. Boxplot (with medians represented by middle bars, interquartile ranges between 25th
and 75th percentiles represented by boxes, and the most extreme data points not considered outliers
represented by whiskers) for biases (%) for the profiles over both urban and non-urban regions
(yellow), profiles over urban regions (green), and profiles over non-urban regions (red) at 600-hPa
layer (panel a), 800-hPa layer (panel b), and the surface layer (panel c). An outlier is a value that
is more than 1.5 times the interquartile range away from the top or bottom of the box.

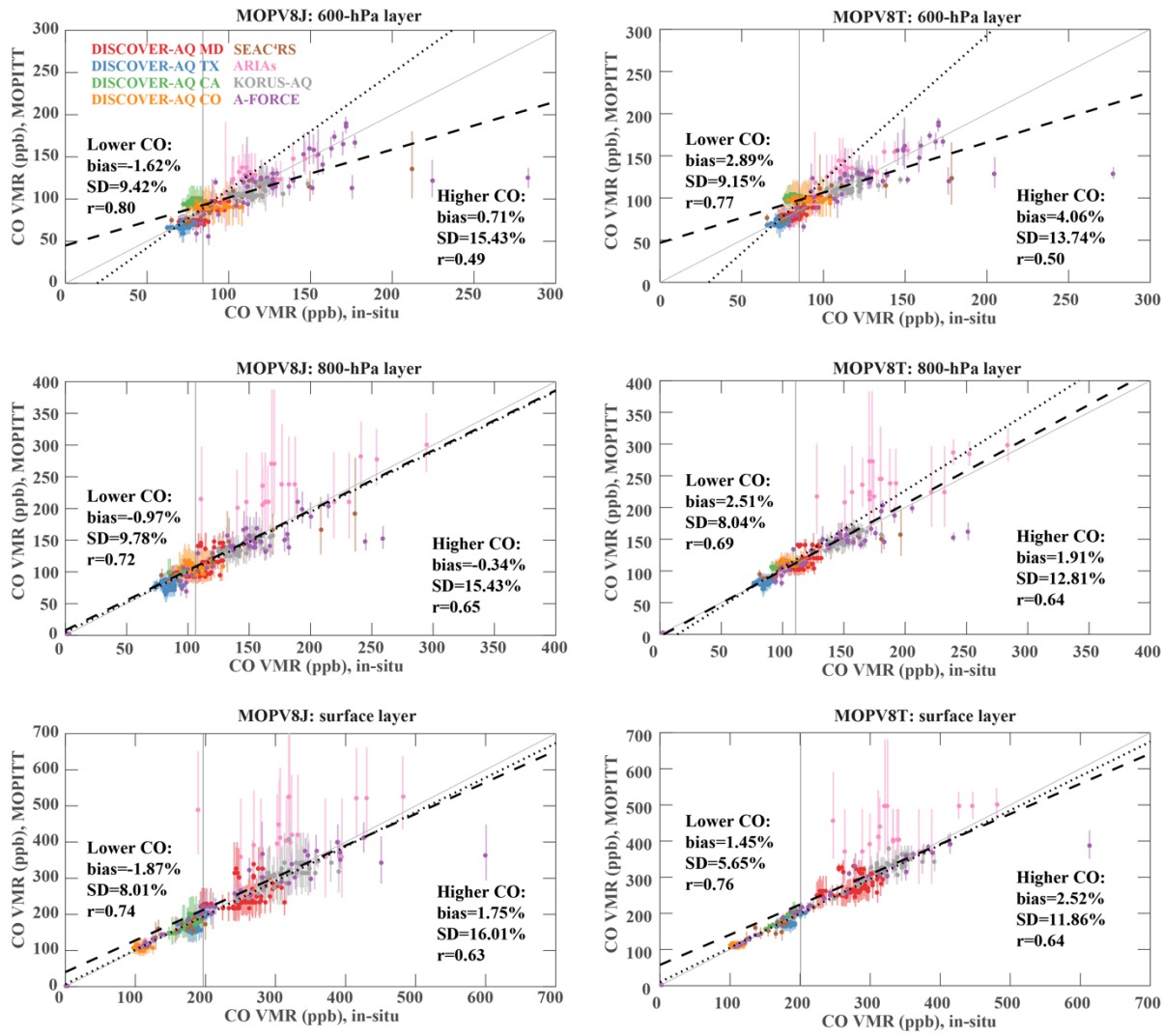


**Figure 7**. MOPITT V8J and V8T validation results at 600-hPa layer, 800-hPa layer, and the surface layer against the lower 50% in-situ profiles of CO and higher 50% in-situ profiles of CO. The variability of the MOPITT data used to calculate each of the plotted mean values are represented by the vertical error bars. Each panel shows the least-squares best-fit lines for the lower 50% CO concentrations (dotted line) and the higher 50% CO concentrations (dashed line).


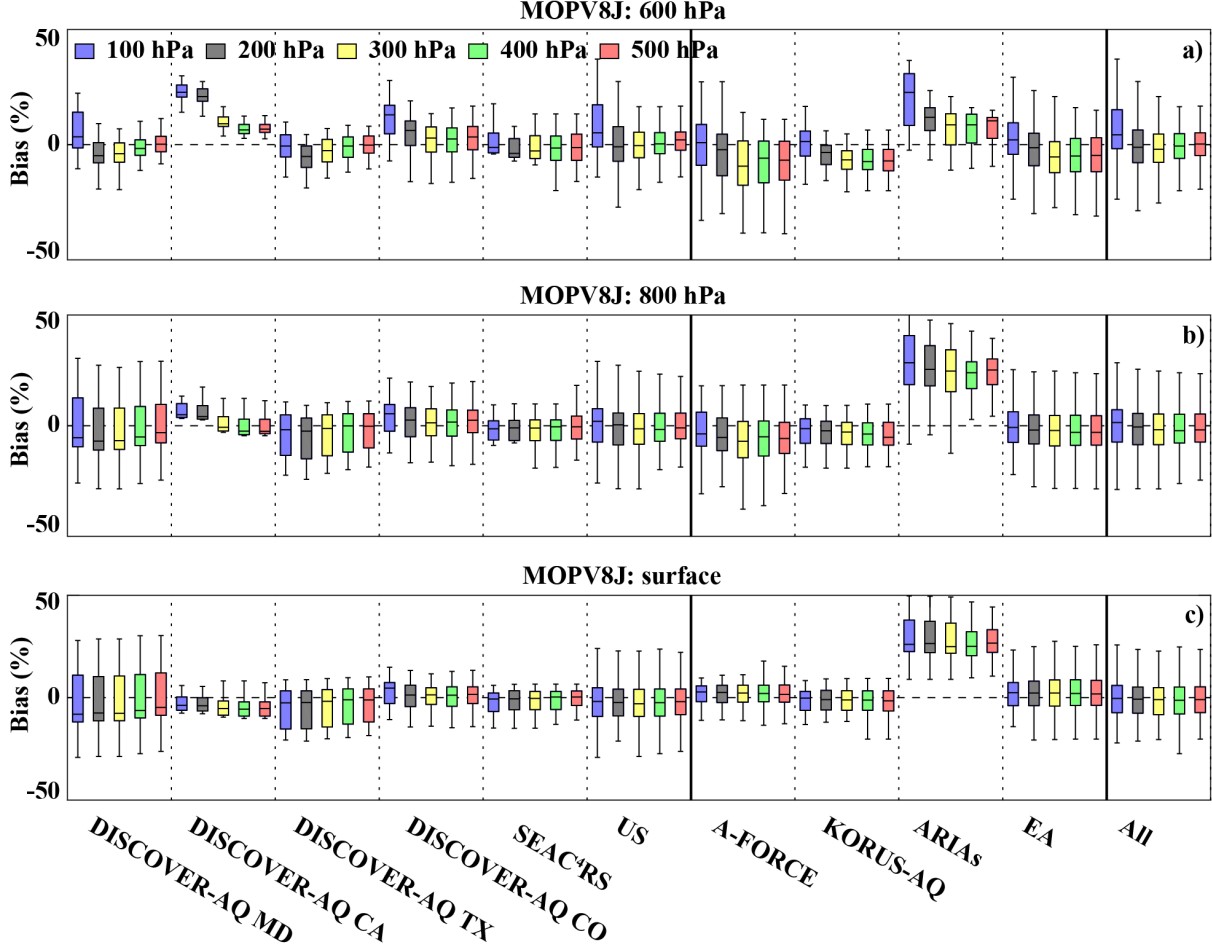

**Figure 8**. Sensitivity to $P_{interp}$. Biases (%) using 100 hPa (blue), 200 hPa (gray), 300 hPa (yellow),
400 hPa (green), and 500 hPa (red) as $P_{interp}$ at 600-hPa layer (panel a), 800-hPa layer (panel b),
and the surface layer (panel c) are shown by boxplot (with medians represented by middle bars,
interquartile ranges between 25th and 75th percentiles represented by boxes, and the most extreme
data points not considered outliers represented by whiskers). The biases are calculated against all
(both urban and non-urban) in-situ profiles listed in Table 1. The "200 hPa" values (gray) in are
the same as yellow values (for all data) in Figure 6. See the caption of Figure 6 for the definition
of outliers.

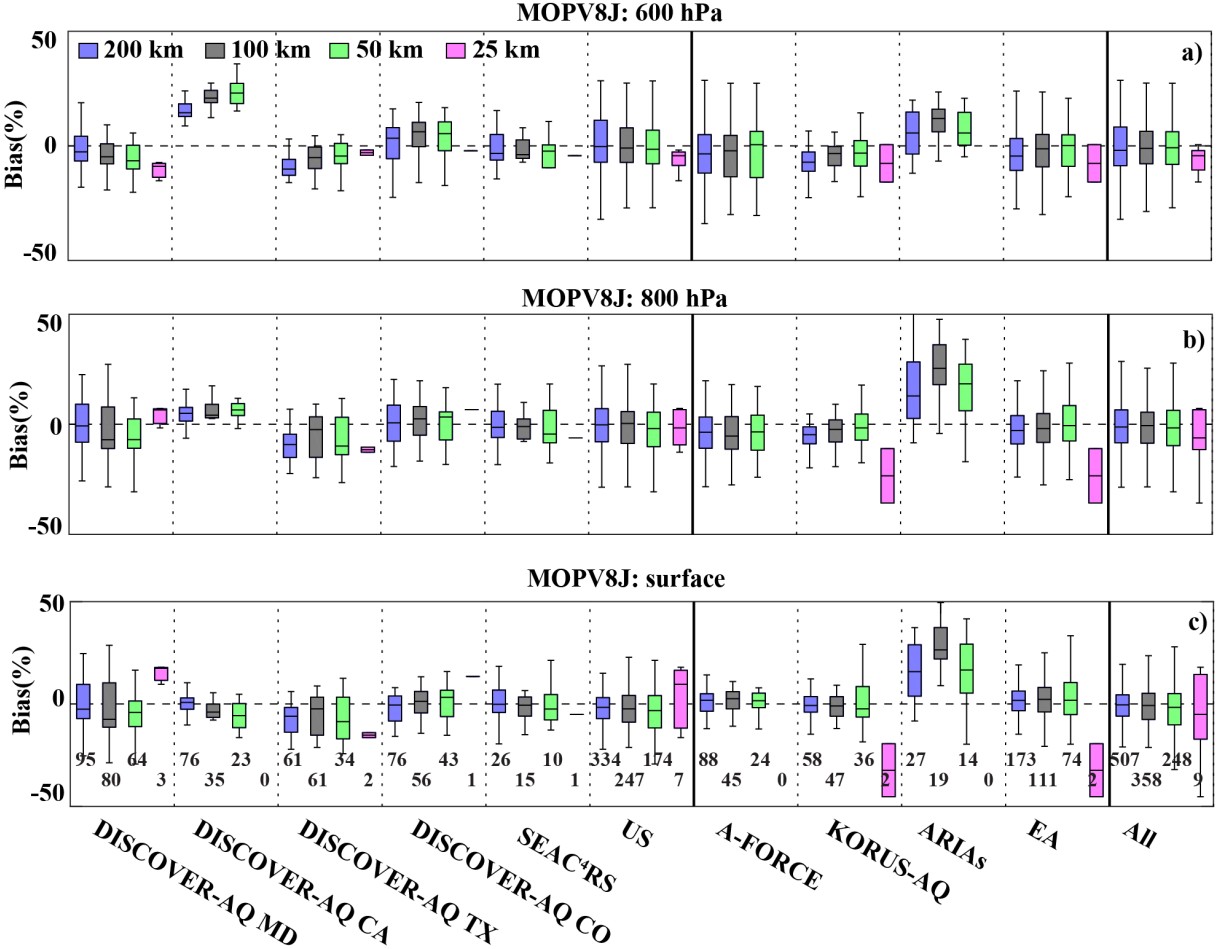

**Figure 9**. Sensitivity to the radius as criteria for co-location. Biases (%) using 200 km (blue), 100
km (gray), 50 km (green), and 25 km (pink) as the radius for co-location at 600-hPa layer (panel
a), 800-hPa layer (panel b), and the surface layer (panel c) are shown by boxplot (with medians
represented by middle bars, interquartile ranges between 25th and 75th percentiles represented by
boxes, and the most extreme data points not considered outliers represented by whiskers). The
numbers in panel c correspond to the number of in-situ profiles qualified for validation within the
given radius. The biases are calculated against all (both urban and non-urban) in-situ profiles listed
in Table 1. The "100 km" values (gray) are the same as yellow values (for all data) in Figure 6.
See the caption of Figure 6 for the definition of outliers.

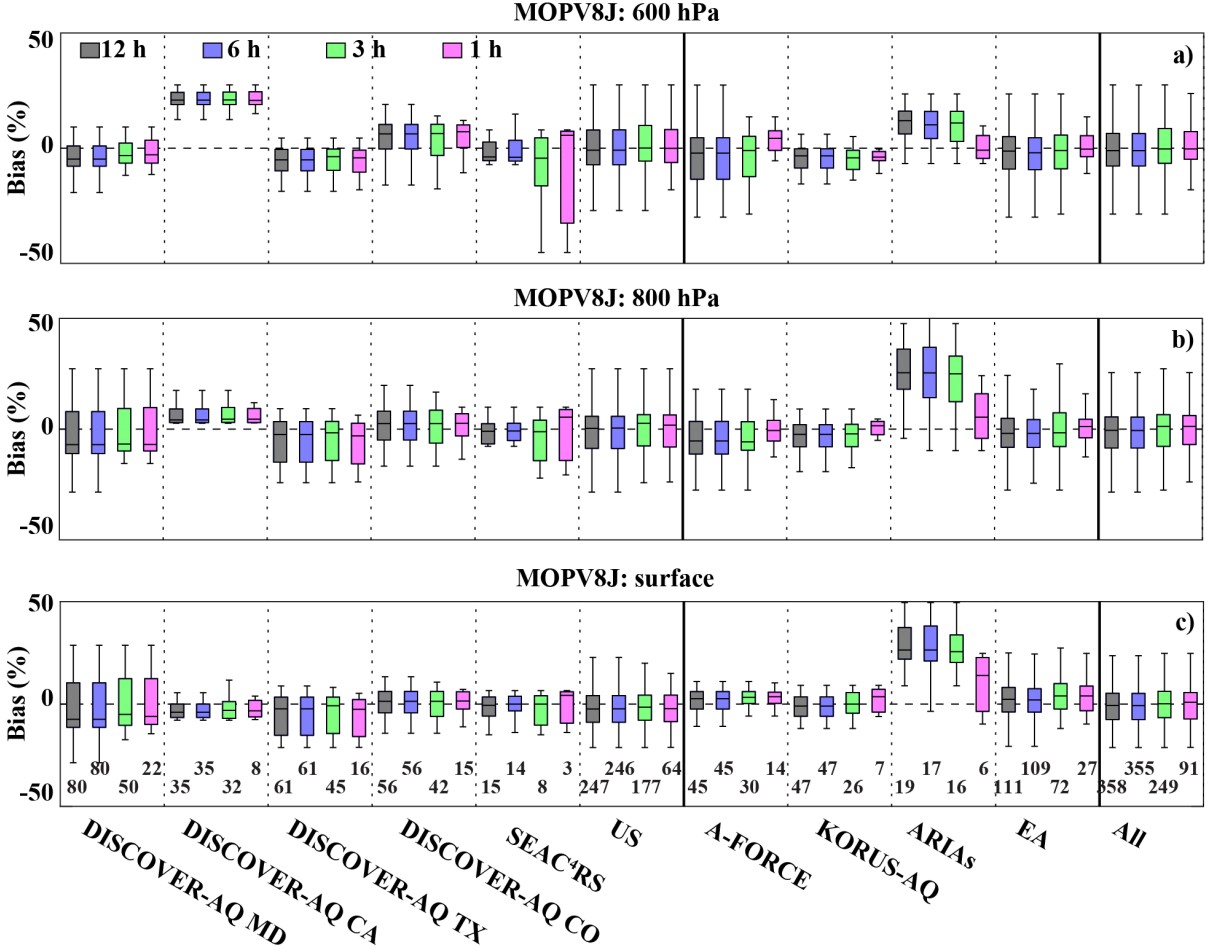

**Figure 10**. Sensitivity to the allowed maximum time difference as criteria for co-location. Biases (%) using 12 hour (gray), 6 hour (blue), 3 hour (green), and 1 hour (pink) as the allowed maximum time difference for co-location at 600-hPa layer (panel a), 800-hPa layer (panel b), and the surface layer (panel c) are shown by boxplot (with medians represented by middle bars, interquartile ranges between 25th and 75th percentiles represented by boxes, and the most extreme data points not considered outliers represented by whiskers). The numbers in panel c correspond to the number of in-situ profiles qualified for validation within the given allowed maximum time difference. The biases are calculated against all (both urban and non-urban) in-situ profiles listed in Table 1. the "12 h" values (gray) are the same as yellow values (for all data) in Figure 6. See the caption of Figure 6 for the definition of outliers.

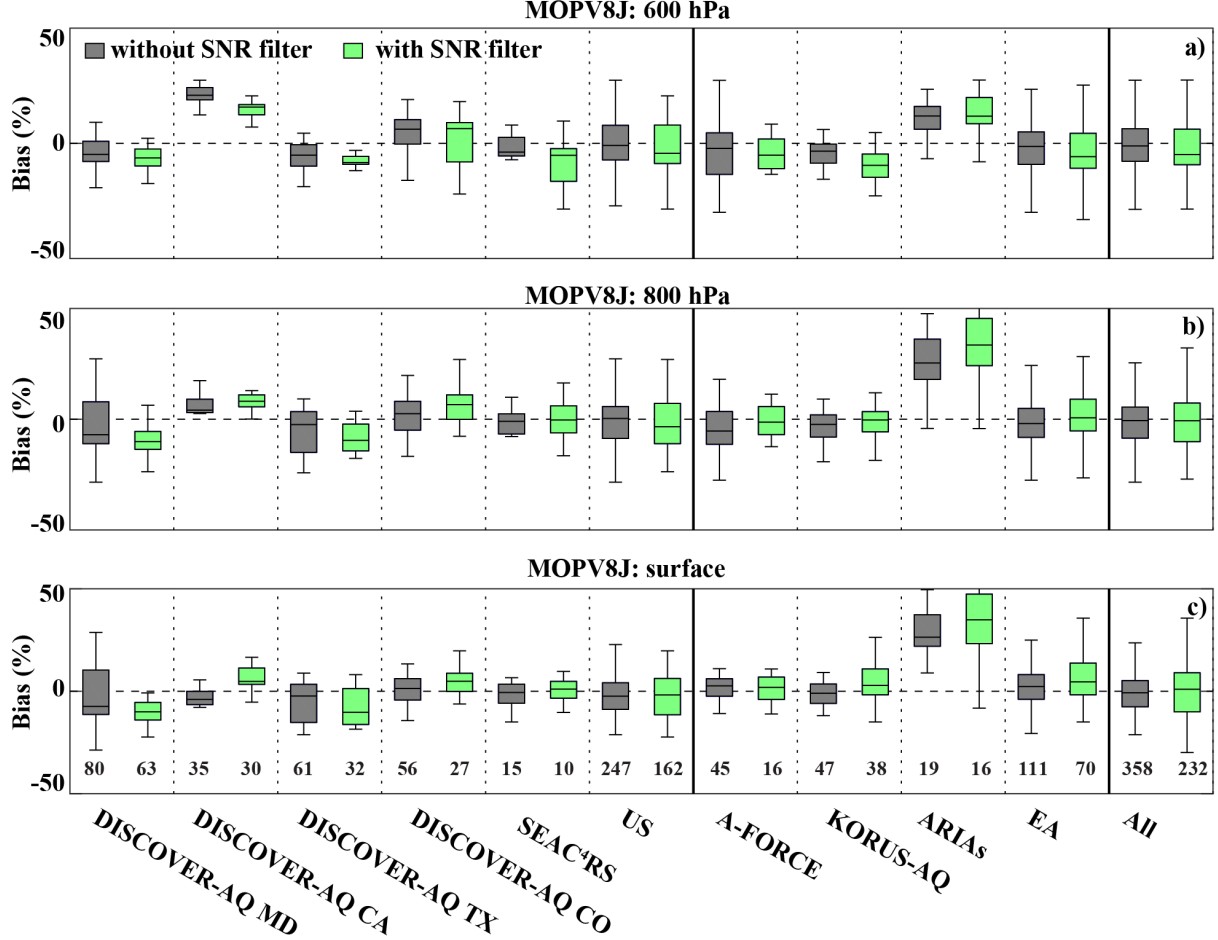

**Figure 11**. Sensitivity to the signal-to-noise ratio (SNR) filters. Biases (%) for MOPITT retrievals
without SNR filters (gray), and MOPITT retrievals with SNR filters (green) at 600-hPa layer
(panel a), 800-hPa layer (panel b), and the surface layer (panel c) are shown by boxplot (with
medians represented by middle bars, interquartile ranges between 25th and 75th percentiles
represented by boxes, and the most extreme data points not considered outliers represented by
whiskers). The numbers in panel c correspond to the number of in-situ profiles qualified for
validation without or with SNR filters. The biases are calculated against all (both urban and non-
urban) in-situ profiles listed in Table 1. the "without SNR filter" values in are the same as yellow
values (for all data) in Figure 6. See the caption of Figure 6 for the definition of outliers.