# Peer review of "Assessing MOPITT carbon monoxide retrievals over urban versus non-urban regions"

_Atmospheric Measurement Techniques, 2019_

## Referee Comment (RC1) · Anonymous Referee #1 · 28 Dec 2019

The focus of this paper is a comparison of the latest MOPITT CO V8 retrievals with aircraft measurements. The authors use aircraft profiles that have not been previously used for comparisons with MOPITT, including profiles around urban regions. A variety of sensitivity tests are also performed to see how specific parameters affect the comparison. Overall I think the paper is well written and sound, and though I have numerous comments they are mostly minor.

General comments

G1: Even though the word "validation" or a variant has been used frequently in these contexts in the literature, "comparison" would nearly always be a better choice. "Validation" has a positive connotation which to me makes it sound like nothing new can be learned, and like the result is already known before the study. However, there is almost

always something new that can be learned and new ways to improve the retrievals. In addition the aircraft measurements themselves are not perfect as the authors point out, with limited measurement altitudes and possible inhomogeneities. Please try to limit use of the word "validation" or a variant to five or fewer instances throughout the entire paper. I include some suggestions for rephrasing in the technical comments, but I do not have a technical comment for each instance.

G2: Sometimes it is not always clear when all data from the listed flights are used in a comparison and when it is just urban areas like the title implies. Please clarify throughout (especially Sect. 4) if comparisons are from just urban or if they include both urban and non-urban.

Specific comments:

S1 - p1l1: Suggest the title be changed to be more descriptive, e.g., "Comparisons of MOPITT carbon monoxide retrievals with aircraft measurements, focusing on urban regions"

S2 - p1l21: list date range of campaigns (2011-2016)

S3 - p1l22: Please specify biases here refer to both urban and non-urban.

S4 - p1l22: Why is V8N disregarded in the abstract? Low DFS?

S5 - p2l30: specify the other levels

S6 - p2l48: Surely MOPITT itself is not doing the CO retrievals, but rather a team at NCAR? Suggest updating to: "Observations from the Measurements…satellite have been used for retrieving…"

S7 - p2l51-52: Similar to last comment, I think MOPITT just makes the measurement and NCAR provides the product. Suggest update to: "…products, a multispectral TIR-NIR product is also produced, which…

S8 - p4l109: I think it would help to update to "quantities in the state vector."

S9 - p5l142: Similar to S6 & S7 – "There are 121 profiles over four urban regions from DISCOVER-AQ."

S10 - p5l145: "...campaign obtained 45 profiles in total sampled over..."

S11 - p6l161: I'm curious, why not just take the aircraft data as high as it goes and then use the model for the rest? Why include more interpolation with a Pinterp parameter than needed?

S12 - p6l169: This is a little circular, comparing MOPITT retrievals with data that assimilated another version of MOPITT retrievals. It would be helpful to let readers know here that you do not compare with these higher levels later, and that they are expected to have a minimal impact on the lower levels you use in the comparison.

S13 - p6l173: Is this mass weighted? If not add in "unweighted averaging"

S14 – p7l176: Please clarify, are these MOPITT profiles with the center point in the radius, or the entire footprint in the radius?

S15 - p7l181-183: Please clarify that these are not the profiles, but rather the state vectors. (You could also remind readers the state vectors and profiles are related by log10).

S16 – p7l197: Specify, what does "uniformly weighted" mean here? In pressure? For MOPITT, isn't the surface level an exception to "uniformly"?

S17 – p7l198: "vertical and horizontal" here is a little confusing (at first I thought it was in km, but realized it is variation in CO). You could reword to "The standard deviation of the original aircraft CO observations in each MOPITT layer are also shown, which is due to horizontal and vertical variability in CO."

S18 - p7l201: Numerically, what is the xtransformed and xrtv difference?

S19 - p8l207: Even if it was not the focus, MOPITT has been compared against other observing systems in urban regions prior to this paper. For example, Buchholz (doi:

10.5194/amt-10-1927-2017) compared MOPITT observations with ground-based observations in urban areas including Toronto and Bremen.

S20 - p9l238: Does this range of percentages include the 900 hPa and 700 hPa layers not shown? Why did you decide not to show these layers? Presumably you already did most the work for their comparison too.

S21 - p9l239: Why wasn't V8N also included? Low DFS? Coverage over land only?

S22 – p9l239: Consider changing "lower" to "smaller." I initially interpreted "lower" to mean "less than" (or more negative), but I think you mean "closer to zero." Same comment for "higher" on line 241.

S23 - p9l240: Is "-0.2" supposed to be "-2.0" based on Table 2?

S24 – p9l242: It looks like you can omit "generally" here, as it appears to be true for all (unless 700 hPa and 900 hPa are exceptions).

S25 - p9l244-246: This line is hard to read because of the number density, and the information is already in Table 2. I suggest omitting it completely.

S26 – p9l247: If you specified for T and J, then you could remove "generally" on this line.

S27 – p9l259: I suggest "at 600 hPa" -> "at the 600-500 hPa layer" (same for line 261). Otherwise it sounds like the comparison is at a specific level.

S28 – p11l295: Consider rewording "this validation of MOPITT at higher CO concentrations..." which sounds like it is the process does not work as well, rather than the results being further off.

S29 – p12l335: Quantitatively how much larger are the "larger biases"?

S30 – p12l339: This is repetitive with a sentence a few lines up (line 329). Can you just separate out a paragraph for DISCOVER-AQ CA so you do not have to mention it

6 times?

S31 - p12l344: "Previous MOPITT evaluation results," are these previous studies? Could you cite a few examples?

S32 - p13l376: Does this section and 4.4 use both urban and non-urban observations?

S33 - p13l380: Please be quantitative with "long enough lifetime" and include a reference.

S34 - p14l394: What is the L3 grid size?

S35 – p15l422: "MOPITT biases" -> "MOPITT mean biases"

S36 – p15l422: Please provide a reference for "10% required accuracy"

S37 – p16l453: Do you know about how many profiles go into each grid cell for Level 3? If it's 1x1 degrees then a 100 km radius is larger. In this case the overall agreement may actually be worsened further by too few MOPITT soundings (if this is what you mean by "which is unlikely to happen when generating L3").

S38 – p16l463-468: These 2 sentences are very late in the paper. They should be earlier, like in Section 2.2.

S39 – p16l468-476: This discussion on NO2 variability from GeoTASO to try to constrain CO variability seems irrelevant and late in the paper. I think the whole thing should be omitted.

S40 – p17l480: All the references to data should be split out into a "Data availability" section. See "Manuscript Composition" here: https://publications.copernicus.org/for_authors/manuscript_preparation.html.

S41 – p17l480: Best practice is for all data to be in a public repository. If this is not possible, then please provide contact information for how the aircraft data can be obtained including ARIAs and A-FORCE.

S42 – p17l480: Include a last access data with all URLs.

S43 – p17l481-482: These seem like 2 references to the same MOPITT data? Which one should readers use?

S44 – p17l493: An "Author Contribution" section is needed: https://publications.copernicus.org/for_authors/manuscript_preparation.html.

S45 – p24Table1: Could you please include the accuracy of CO from aircraft measurements somewhere?

S46 – p24lTable1: For better traceability, please list the CO scale the aircraft measurements were tied to (e.g., WMO-CO-X2004, WMOCO-X2014, WMO-CO-X2014A, CSIRO...).

S47 – p26: (No response requested) – Figure 1 is well done and has high information content

S48 – p27l776-777: I don't understand "vertical and horizontal variability" here. Does "horizontal" somehow correspond to how many km were flown? Or are these just the standard deviations of aircraft profiles. If so, then just say "are the standard deviations of the original aircraft observations."

S49 – p27l777-778: I guess this is why the black and orange traces do not always match. Optionally consider plotting at layer centers (shifting up by about 50 hPa).

S50 – p29l793: Please define Delta log(VMR) here and explicitly include the base of the logarithm.

S51 – p30l803: Should "Figure 2" be "Figure 4"?

S52 – p31l810: Numerically, what are considered "outliers"? Please also add to captions of Figures 8-11. Or just reference the caption the Figure 6 so it is less repetitive.

S53 – p33lFigure8: Are these MOPITT biases compared with aircraft still? So the "200

hPa" values are the same as yellow values in Figure 6? Please specify or optionally consider showing as a % bias compared with the baseline "200 hPa" results.

S54 – p33lFigure8: Please clarify that you are using "all" observations (both urban and non-urban).

S55 – p34Figure9: Optionally consider comparing against 100 km.

S56 – p35Figure10: Optionally consider comparing against 12 hours.

S57 – Figure S3: It would help to remind readers that "in situ" is a combination of aircraft and models since values at 1050 hPa do not make much physical sense. (As a side observation it is interesting that MOPITT is so insensitive).

S58 – Figure S5: Could you please provide more detail in the caption? Consider marking the level of highest aircraft measurements (presumably this is why there are straight lines).

Technical comments:

T1 - p1l17: "The performance of the" could be omitted (and update has -> have)

T2 - p1l19-20: E.g., validate -> compare, using -> with

T3 - p1l25: suggest "performance" -> "agreement" and adding "with aircraft measurements" after V8T

T4 - p2l32: "allowed maximum" -> "maximum allowed" and "as criteria" -> "criterion"

T5 - p2l34-35: suggest "hence few MOPITT retrievals are included in the comparison."

T6 - p2l36: "overall smaller" -> "smaller overall"

T7 - p2l40: "retrievals that result for comparison." -> "retrievals for the comparison."

T8 - p3l58: "the most recently"

T9 - p3l83-84: suggest "...we compare MOPITT version...regions with aircraft profiles

made over..."

T10 - p3l83: "version" is lowercase here but capitalized on p2l55. Please be consistent throughout.

T11 - p4l100: "retrievals" -> "observations"

T12 - p4l111: "The two" -> "These two"

T13 - p5l124: "determined" -> "considered"

T14 - p5l128-130: move/modify "the profiles over urban and non-urban areas are similar" to right after "We also notice..."

T15 - p5l139: omit "different instruments" (it's implied)

T16 - p6l148: "Only few" -> "Few"

T17 - p6l162: omit "below" (I initially was confused because I thought "below" meant lower pressure/higher altitude)

T18 - p6l174-175: "We have investigated the..."

T19 - p7l179: "have been further" -> "are"

T20 - p7l186: "If fewer than five MOPITT retrievals are co-located with an in-situ profile, the..."

T21 - p7l187-191: I think it would be clearer if you reordered the description. a) In situ profile individually applied to AK and prior of each MOPITT retrieval to get xtranformed. b) xtransform averaged as log10. c) Corresponding MOPITT profile retrievals also averaged

T22 - p7l200: "variability" -> "standard deviation"

T23 - p7l202 & 203: omit "retrieval" (the size does not depend on the retrieval algorithm, but is inherent in the MOPITT observation system)

T24 - p8l205: omit "very"

T25 - p8l207: "validated" -> "compared with aircraft observations"

T26 - p8l209-210: "...show a sensitivity analysis in Section 4 to provide..."

T27 - p8l211: omit "validation"

T28 - p8l215: maybe "validation" -> "comparison with aircraft profiles"

T29 - p8l224: "against observations" -> "against in situ observations"

T30 - p9l236: "Corresponding results" -> "These comparisons"

T31 - p9l244: "three levels" -> "three levels in Table 2"

T32 – p9l251: "...in terms of higher correlation coefficients and smaller biases..."

T33 - p9l253: "provide" -> "evaluate", "evaluation against" -> "retrievals during", "campaigns" -> "campaigns with results"

T34 - p9l264: "in more favorable weather conditions" -> "during times with greater vertical mixing"

T35 - p9l271: "xin-situ, the" -> "xin-situ over non-urban areas, the"

T36 - p9l291: "concentrations all" -> "concentration for all"

T37 - p9l292: "For both" -> "For the higher 50% of measured mixing ratios both" and omit "if only the upper 50% of measured mixing ratios are considered"

T38 - p12l335-336: Suggest omitting "The validation results", and the second "validation" and changing "are" -> "does", "different for" -> change

T39 - p12l347: "the radius" -> "a radius"

T40 - p12l350: "close" -> "similar"

T41 - p13l356: "a smaller number of included" -> "including fewer"

T42 - p13l359: "a a more more" -> "a more"

T43 - p13l367: "especially" -> "including"

T44 - p14l399: "Level 3" -> "the Level 3"

T45 - p14l411: omit "process"

T46 – p15l423: suggest "overall" -> "mean"

T47 – p15l424: "to 3.5% for different levels"

T48 – p15l429: "to" -> "into"

T49 – p15l431: "compared with low"

T50 – p15l440: "as co-location criteria" -> "as a co-location criterion"

T51 – p15l441: "where a" -> "where only a"
* * *

---

## Referee Comment (RC2) · Anonymous Referee #2 · 31 Dec 2019

This manuscript by Wenfu Tang et al presented a comparison of the latest MOPITT CO V8 retrievals with aircraft measurements from DISCOVER-AQ, SEAC4RS, ARIAs, A-FORCE, and KORUS-AQ campaigns conducted over the US or East Asia. In addition, the sensitivities of validation results to assumptions and data filters applied during the comparisons of MOPITT retrievals and in-situ profiles were also performed and analyzed. The comparison between the MOPITT CO product with various version and the coincident observations has been previously performed by many scientists in many groups around the world. This study is an extension of previous study and the strategy for comparison has been used extensively in previous MOPITT evaluation and validation studies. However, this study is one of few studies that focus on comparison over around urban regions, this is interesting. Overall, this paper is well written and fits well

within the scope of AMT. I recommend for publication though I rate the novelty of this paper as moderate. Since referee # 1 has listed numerous technical comments which are mostly overlapped with my comments. Here I don't present the repeated correction request. Extra minor revisions or comments are: 1. The Base map and color bar in Fig. 1 can be improved. It is hard to distinguish one from another. In latitude and longitude axis, the number like 30 should be 30°. 2. What does the dashed line in Figs. 4 and 5 mean? The one to one lineïij§Should be stated in the caption. 3. If you only compare the results at surface, 800 hPa, and 600 hPa. Then the expression should be the concentrations at surface, 800 hPa, and 600 hPa rather than the profiles at surface, 800 hPa, and 600 hPa. Another confusing thing is that the MOPITT could have a very low DOFS at a given level with a limited range (Fig. 3). Thus, the retrieval should come more from a priori information rather than the measurement. In other words, I guess, the good agreement between the MOPITT and aircraft at a given level is largely attributed to the a priori information and the smoothing effect in equation 2.

---

## Referee Comment (RC3) · Anonymous Referee #3 · 10 Jan 2020

Review of

Validation of MOPITT Carbone Monoxide (CO) retrievals over urban regions

By Tang et al.

Manuscript: amt-2019-419

General comments:

The aim of this paper is to evaluate two versions of MOPITT CO (V7 and V8) by comparison with aircraft observations from diverse campaigns all over the globe. Each version has two sub versions (V7-8T, V7-8N, V7-8J for thermal, NIR and TIR+NIR, respectively). Urban and non urban areas are the focus of the evaluation. This is a

paper that complete the list of publications of the evaluation of the different versions of MOPITT CO. Lots of statistics are provided and the MOPITT users community could find some interest in order to interpret MOPITT data over urban areas.

However, I found the comparison sometimes difficult to follow because of the large number of campaigns, the number of aircraft profiles by campaign, number of aircraft profiles over urban regions, the number of MOPITT CO profiles in different circles,.. Table 1 helps but if possible it would be nice to simplify in the text. Moreover, the title does not reflect totally the subject of the paper: the validation of the MOPITT CO retrievals is also over non urban regions. I suggest to change the title in that way.

1) Moreover, the distinction of urban and non urban regions for the comparison of MO- PITT CO with aircraft observations could mislead the reader. What is important in this study, is it the carbone monoxide emitted from the urban region or just the urban region with surface parameters different from non urban regions? Such surface parameters that are used in the retrievals of MOPITT CO (surface temperature, emissivity). At 600 hPa, some comparisons are done but this is above the boundary layer. There is a great chance that the CO measured by both MOPITT and the aircraft is transported from other regions that are not representative of urban regions. The author should clarify this point.

2) Also, it would be nice to have a clear recommandation on which MOPITT CO version to use. For example, after reading the table 2 of the paper, I found difficult to conclude on which version to use for urban or a non urban study as well. The statistics are often very similar and I was wondering what is the added value of V8 vs V7 and how significant the values are? It would be nice the authors discuss this point and conclude with clear recommendation in the conclusions on the use of the different versions of MOPITT CO.

3) The Section 4.4 (Sensitivity to the signal-to-noise ratio (SNR) filters) is unclear to me. What are the conclusions we can draw from this section? Is level 3 useless? I

didn't catch the point of this section. Maybe the authors could clarify on how to use Level 3 data over urban and non urban regions in the light of of the use of such SNR filter.

Specific Comments:

Abstract:

In the paper, V7 and V8 of MOPITT CO are evaluated whereas only V8 is mentioned in the abstract.

Section 3.3

L 300-301: This means MOPITT CO concentrations are highly variable in circles where true concentrations are high. In this condition, what are the retrieval errors for these MOPITT pixels?

L 334: please correct the sentence

L 360: please correct the sentence

L 369: The sentence 'we note..' is unclear to me. Please clarify if necessary.

---

## Author Comment (AC1) · 31 Jan 2020

The focus of this paper is a comparison of the latest MOPITT CO V8 retrievals with aircraft measurements. The authors use aircraft profiles that have not been previously used for comparisons with MOPITT, including profiles around urban regions. A variety of sensitivity tests are also performed to see how specific parameters affect the comparison. Overall I think the paper is well written and sound, and though I have numerous comments they are mostly minor.
Response: Thank you for your time and effort in reviewing our manuscript.

General comments
G1: Even though the word "validation" or a variant has been used frequently in these contexts in the literature, "comparison" would nearly always be a better choice. "Validation" has a positive connotation which to me makes it sound like nothing new can be learned, and like the result is already known before the study. However, there is almost always something new that can be learned and new ways to improve the retrievals. In addition the aircraft measurements themselves are not perfect as the authors point out, with limited measurement altitudes and possible inhomogeneities. Please try to limit use of the word "validation" or a variant to five or fewer instances throughout the entire paper. I include some suggestions for rephrasing in the technical comments, but I do not have a technical comment for each instance.
Response: We revised the title (see response to the comment #S1) following the reviewer's comment #S1. And by addressing the specific comments and technical comments regarding the use of the term "validation", we have substantially reduced the use of the term "validation" and its variant (e.g., "validated", "validating"). Please see responses to the comments # S28, T2, T25, T27, T28, T38. In addition, we also tried to use terms such as "agreement" and "comparison" to replace "validation" when applicable, in our description and discussion of our own results in the manuscript. Note that we did not change the usage of "validation" when we refer to some previous studies because "validation" and "evaluation" are more effective to describe these studies than a substitute, for example "*the primary goal of DISCOVER-AQ was to provide aircraft observation methodologies for satellite validation*". Please see the updated manuscript for details.

G2: Sometimes it is not always clear when all data from the listed flights are used in a comparison and when it is just urban areas like the title implies. Please clarify throughout (especially Sect. 4) if comparisons are from just urban or if they include both urban and non-urban.
Response: Thank you for pointing this out. In Section 3 we analyze "all profiles", "profiles over urban regions", and "profiles over non-urban regions". While in Section 4, we use all the in-situ profiles listed in the Table 1 regardless of if the profiles are over urban or non-urban regions. We explicitly added the following statement at the beginning of Section 4:
"*While in Section 3 we compared profiles over urban regions, and profiles over non-urban regions separately to MOPITT V8T, V8N, V8J, V7T, V7N, and V7J, in this section, we compare only the MOPITT V8J product to all the in-situ profiles (both over urban and non-urban regions) described in Table 1 to test the sensitivity of comparison results to the assumptions made during the comparisons.*"
We separated Section 3.2 (Discussions on individual campaigns) into 5 paragraphs so that each paragraph discusses overall conclusions or comparisons with one specific campaign.
In addition, we explicitly listed what campaign data were used for each statement in Section 4.

Specific comments:

S1 - p1| 1: Suggest the title be changed to be more descriptive, e.g., "Comparisons of MOPITT carbon monoxide retrievals with aircraft measurements, focusing on urban regions"

Response: We changed the manuscript title to

"Assessing MOPITT carbon monoxide retrievals over urban versus non-urban regions".

S2 - p1| 21: list date range of campaigns (2011-2016)

Response: We added date range for the campaigns "*DISCOVER-AQ (2011-2014), SEAC4RS (2013), ARIAs (2016), A-FORCE (2009, 2013), and KORUS-AQ (2016)*".

S3 - p1| 22: Please specify biases here refer to both urban and non-urban.

Response: We changed the sentence "*Overall, MOPITT performs reasonably well over both urban and non-urban regions, overall biases for V8J and V8T vary from -0.7% to 0.0%, and from 2.0% to 3.5%, respectively.*"

to

"*In general, MOPITT agrees reasonably well with the in-situ profiles, over both urban and non-urban regions. Version 8 multispectral product (V8J) biases vary from -0.7% to 0.0% and version 8 thermal-infrared product (TIR) biases vary from 2.0% to 3.5%.*"

S4 - p1| 22: Why is V8N disregarded in the abstract? Low DFS?

Response: Thank you for pointing this out. MOPITT V8N product has relatively lower DFS (Figure 3), and are not as widely used as the V8J and V8T. Therefore, V8N is not the main focus of this study and it is only analyzed in Table 2.

S5 - p2| 30: specify the other levels

Response: We added "(e.g., 600 hPa)" in the sentence.

S6 - p2| 48: Surely MOPITT itself is not doing the CO retrievals, but rather a team at NCAR? Suggest updating to: "Observations from the Measurements…satellite have been used for retrieving…"

Response: We updated the sentence accordingly.

S7 - p2| 51-52: Similar to last comment, I think MOPITT just makes the measurement and NCAR provides the product. Suggest update to: …products, a multispectral TIRNIR product is also produced, which…

Response: We changed "*MOPITT also provides the multispectral TIR-NIR product*" to "*the MOPITT multispectral TIR-NIR product is also provided*".

S8 - p4| 109: I think it would help to update to "quantities in the state vector."

Response: We updated the sentence accordingly.

S9 - p5| 142: Similar to S6 & S7 – "There are 121 profiles over four urban regions from DISCOVER-AQ."

Response: We updated the sentence accordingly.

S10 - p5| 145: "… campaign obtained 45 profiles in total sampled over..."

Response: We revised the sentence.

S11 - p6| 161: I'm curious, why not just take the aircraft data as high as it goes and then use the model for the rest? Why include more interpolation with a Pinterp parameter than needed?
Response: Thank you. We did take the aircraft data as high as it goes, and use the reanalysis data for the rest. $P_{interp}$ is used because the rest of the pressure levels need to be filled differently. For pressure levels below $P_{interp}$ (lower altitude), values are linearly interpolated using highest-altitude aircraft measurement and reanalysis data at $P_{interp}$. For pressure levels above $P_{interp}$ (higher altitude), reanalysis data are used directly. If we use reanalysis data to fill all the levels directly, the extended vertical profile may not be continuous at the highest-altitude the aircraft profile. In addition, the use of $P_{interp}$ allows us to test the sensitivity to the use of model or reanalysis data as because parameter $P_{interp}$ controls the impact of the model-based profile extension on the shape and value of in-situ profiles (see Figure S5).

S12 - p6| 169: This is a little circular, comparing MOPITT retrievals with data that assimilated another version of MOPITT retrievals. It would be helpful to let readers know here that you do not compare with these higher levels later, and that they are expected to have a minimal impact on the lower levels you use in the comparison.
Response: We thank the reviewer for pointing this out. We added the following statement in the text:
*"We note that as we do not compare with these higher levels later, the use of CAMS reanalysis is expected to have a minimal impact on the lower levels we use in the comparison (e.g., surface, 800 hPa, and 600 hPa)."*

S13 - p6| 173: Is this mass weighted? If not add in "unweighted averaging"
Response: We added the adjective "unweighted" before "averaging".

S14 – p7| 176: Please clarify, are these MOPITT profiles with the center point in the radius, or the entire footprint in the radius?
Response: Thank you. It is the center point that needs to be in the radius. We revised the sentence to:
*"MOPITT profiles are considered co-located with the aircraft profile and are selected for comparison only if their center points are within the radius of 100 km and within 12 hours of the acquisition of the aircraft profile."*

S15 - p7| 181-183: Please clarify that these are not the profiles, but rather the state vectors. (You could also remind readers the state vectors and profiles are related by log10).
Response: As shown in the following revised text, we have re-emphasized that all CO profiles ($x_{in-situ}$, $x_a$, and $x_{transformed}$) appearing in Equation (2) are expressed in terms of $\log_{10}(VMR)$.
*"For each pair of co-located MOPITT retrieved and in-situ profiles, we apply the MOPITT a priori profile and averaging kernel to the in-situ profile as in Eq. (1). Thus, after converting from profiles of the in-situ and a priori CO concentrations to $\log_{10}(VMR)$ profiles ($x_{in-situ}$ and $x_a$,), we calculate*

$$x_{transformed} = x_a + A(x_{in-situ} - x_a) \qquad (2)$$

*so that the $\log_{10}(VMR)$-based transformed in-situ profile ($x_{transformed}$) has the same degree of*

*smoothing and a priori dependence as the MOPITT retrieved log$_{10}$(VMR) profile ($x_{rtv}$)."*

S16 – p7| 197: Specify, what does "uniformly weighted" mean here? In pressure? For MOPITT, isn't the surface level an exception to "uniformly"?
Response: "uniformly weighted" is about the way that the MOPITT retrieval algorithm internally converts from 'retrieval-grid' CO profiles (10 levels/layers) to higher-resolution 'model-grid' CO profiles (35 levels/layers) that are needed for the radiative transfer model. For V5 and later products, the algorithm assumes uniform weighting when it makes this conversion. So, for example, the retrieved CO at 900 hPa represents a layer from 900 to 800 hPa with a constant VMR within that layer. Internally, this means that model levels at 900, 875 and 850 are all assigned the same VMR value. We notice "uniformly weighted" is not relevant in this sentence and hence changed it to "uniform".

S17 – p7| 198: "vertical and horizontal" here is a little confusing (at first I thought it was in km, but realized it is variation in CO). You could reword to "The standard deviation of the original aircraft CO observations in each MOPITT layer are also shown, which is due to horizontal and vertical variability in CO."
Response: We revised the sentence accordingly.

S18 - p7| 201: Numerically, what is the xtransformed and xrtv difference?
Response: We added the numerical value (12.4 ppb) in the sentence.

S19 - p8| 207: Even if it was not the focus, MOPITT has been compared against other observing systems in urban regions prior to this paper. For example, Buchholz (doi: 10.5194/amt-10-1927-2017) compared MOPITT observations with ground-based observations in urban areas including Toronto and Bremen.
Response: Thank you for pointing this out. We added a statement regarding the comparisons with ground-based spectrometric column retrievals, and citation for Buchholz et al. (2017) and Hedelius et al. (2019), in the introduction:
*"In addition, MOPITT products have also been compared with ground-based spectrometric column retrievals (e.g., Buchholz et al., 2017; Hedelius et al., 2019)."*
And for the sentence pointed out by the reviewer *"... that MOPITT has yet to be validated over urban regions"*, we added specific description *"... that MOPITT has yet to be validated over urban regions with in-situ observations."*

S20 - p9| 238: Does this range of percentages include the 900 hPa and 700 hPa layers not shown? Why did you decide not to show these layers? Presumably you already did most the work for their comparison too.
Response: The range of percentages only includes what is shown in the Table 1 (i.e., surface layer, 800 hPa, and 600 hPa). We present results on surface layer, 800 hPa, and 600 hPa in this study to represent three levels, namely the surface level, the top of the PBL (800 hPa), and the free troposphere (600 hPa). We do not show results for 900 hPa and 700 hPa as surface layer, 800 hPa, and 600 hPa are representative, as shown by the in-situ profiles (Figures 2, S1, S3, S4) and the averaging kernels at those layers (Figure 3).

S21 - p9| 239: Why wasn't V8N also included? Low DFS? Coverage over land only?

Response: Please see the reply to the comment S4.

S22 – p9| 239: Consider changing "lower" to "smaller." I initially interpreted "lower" to mean "less than" (or more negative), but I think you mean "closer to zero." Same comment for "higher" on line 241.
Response: We changed "lower" to "smaller", and "higher" to "larger".

S23 - p9| 240: Is "-0.2" supposed to be "-2.0" based on Table 2?
Response: We thank the reviewer for noticing this. We changed "*from -0.2% to -0.8%*" to "*from -0.8% to -2%*"

S24 – p9| 242: It looks like you can omit "generally" here, as it appears to be true for all (unless 700 hPa and 900 hPa are exceptions).
Response: We deleted "generally".

S25 - p9| 244-246: This line is hard to read because of the number density, and the information is already in Table 2. I suggest omitting it completely.
Response: We removed the sentence "*For example, for the V8J product, correlation coefficients over urban regions are 0.53, 0.57, and 0.53 at the surface, 800 hPa, and 600 hPa, respectively, while over non-urban regions, the corresponding correlation coefficients are 0.76, 0.73 and 0.67.*"

S26 – p9| 247: If you specified for T and J, then you could remove "generally" on this line.
Response: We changed the sentence "*We also notice that V8 products generally have higher correlation coefficients with in-situ measurements than V7 over non-urban regions, whereas over urban regions, V8 products generally have lower correlation coefficients than V7.*"
to
"*We also notice that for TIR-NIR and TIR-only products, V8 have higher correlation coefficients with in-situ measurements than V7 over non-urban regions, whereas over urban regions, V8 products have lower correlation coefficients than V7 (except for 600 hPa).*"

S27 – p9| 259: I suggest "at 600 hPa" -> "at the 600-500 hPa layer" (same for line 261). Otherwise it sounds like the comparison is at a specific level.
Response: We changed "at 600 hPa" to "at the 600-hPa layer (i.e., the 100 hPa uniform layer immediately above 600 hPa)", and changed "at the surface" to "at the surface layer (i.e., the uniform layer immediately above the surface)". In addition, we changed 600 hPa, 800 hPa, and the surface to the 600-hPa layer, the 800-hPa layer, and the surface level throughout the paper to indicate the comparisons are not for a specific pressure level.

S28 – p11| 295: Consider rewording "this validation of MOPITT at higher CO concentrations..." which sounds like it is the process does not work as well, rather than the results being further off.
Response: We changed "*this validation of MOPITT at higher CO concentrations*" to "*the agreement between MOPITT and the in-situ profiles at higher CO concentrations*".

S29 – p12| 335: Quantitatively how much larger are the "larger biases"?
Response: We changed the sentence "*The validation results using 100 hPa as Pinterp have larger biases.*" to "*The validation results against DISCOVER-AQ CA using 100 hPa or 200 hPa as*

*Pinterp have larger biases at the 600-hPa layer (~25%).*"

S30 – p12| 339: This is repetitive with a sentence a few lines up (line 329). Can you just separate out a paragraph for DISCOVER-AQ CA so you do not have to mention it 6 times?
Response: We deleted this sentence "*As mentioned in Section 3.2, the DISCOVER-AQ CA aircraft measurements concentrate below 600 hPa, so CO values in the in-situ profiles at 600 hPa and above are filled with and are more sensitive to CAMS reanalysis data.*" Please also see the response to the comment #G2.

S31 - p12| 344: "Previous MOPITT evaluation results," are these previous studies? Could you cite a few examples?
Response: We added Deeter et al. (2012) and Deeter et al. (2016) as examples.

S32 - p13| 376: Does this section and 4.4 use both urban and non-urban observations?
Response: Yes. We added the following statement at the beginning of Section 4:
"*In Section 3, we compared profiles over urban and non-urban regions separately to MOPITT V8T, V8N, V8J, V7T, V7N, and V7J. In this section, we compare only the MOPITT V8J product to all the in-situ profiles (both over urban and non-urban regions) described in Table 1 to test the sensitivity of results to the assumptions made during the comparisons.*"

S33 - p13| 380: Please be quantitative with "long enough lifetime" and include a reference.
Response: The typical lifetime of CO is approximately a month. We added Gamnitzer et al. (2006) as reference.

S34 - p14| 394: What is the L3 grid size?
Response: The resolution is 1°×1°, we added this in the text.

S35 – p15| 422: "MOPITT biases" -> "MOPITT mean biases"
Response: Thank you. We have revised accordingly.

S36 – p15| 422: Please provide a reference for "10% required accuracy"
Response: We added the following reference:
Drummond, J. R., & Mand, G. S. (1996). The Measurements of Pollution in the Troposphere (MOPITT) instrument: Overall performance and calibration requirements. Journal of Atmospheric and Oceanic Technology, 13(2), 314-320.

S37 – p16| 453: Do you know about how many profiles go into each grid cell for Level 3? If it's 1x1 degrees then a 100 km radius is larger. In this case the overall agreement may actually be worsened further by too few MOPITT soundings (if this is what you mean by "which is unlikely to happen when generating L3").
Response: We thank the reviewer for bringing this issue up.
      As described in the MOPITT Version 8 Product User's Guide (https://www2.acom.ucar.edu/sites/default/files/mopitt/v8_users_guide_201812.pdf), MOPITT Level 2 data all feed in the specific filtering rules (both pixel filtering and signal-to-noise ratio). Data Averaging to generate MOPITT Level 3 data is performed on a one-degree latitude/longitude grid (1°×1°). The original reason for implying which is unlikely to happen when generating Level

3 is that the daily MOPITT L2 data size is large, and globally speaking there will be enormous data. However, it is true that 1°×1° pixel size is smaller than 100 km radius. And as the reviewer points out, for each individual 1°×1° grid, we should not expect to have much more data to perform the filter and averaging. So we have deleted the statement in the Section 4.4 "However, when generating Level 3 data from Level 2 data, the circumstance is different as there are usually much more data to perform the filter and averaging." We also deleted the statement "which is unlikely to happen when generating Level 3 data" in the Section 5.

However, we disagree that the overall agreement may actually be worsened further by too few MOPITT soundings in MOPITT Level 3 data. We notice that some of the discussions in the manuscript may be misleading. For example, the statement "*In some cases, applying the SNR filters degrades the validation results (e.g., DISCOVER-AQ DC at the surface, DISCOVER-AQ CA at the surface, KORUS-AQ at 600 hPa, and ARIAs at the surface, 800 hPa, and 600 hPa)"* in the section 4.4 to avoid confusion" indicate that applying SNR filter may worsen the agreement in some cases. While this statement is true, it is misleading because the readers may ignore the fact that applying SNR filter also improve the agreement in some cases (e.g., DISCOVER-AQ DC at the 600-hPa layer and the DISCOVER-AQ TX at the 600-hPa layer, and that the overall agreement does not change significantly. Therefore, we delete the aforementioned statement. In addition, we change also changed the statement "*We find that applying the SNR filters does not improve the overall agreement between MOPITT retrievals and the in-situ profiles*" in Section 4.4 to "*We find that applying the SNR filters does not significantly change the overall agreement between MOPITT retrievals and the in-situ profiles used in this study.*" We also changed the statement in the section 5 "*Applying SNR filters does not necessarily improve the overall agreement between MOPITT retrievals and in-situ profiles used in this study*" to "*Applying SNR filters does not necessarily change the overall agreement between MOPITT retrievals and in-situ profiles used in this study significantly*".

In addition, even though applying SNR filter when generating Level 3 data does not significantly change the agreement with the in-situ profiles used in this study, by excluding low-SNR observations from the Level 3 cell-averaged values raises overall mean DFS values (MOPITT Version 8 Product User's Guide, 2018). In addition, the Level 3 product typically are less affected by random retrieval errors (e.g., due to instrument noise or geophysical noise). We have added this statement at the end of section 4.4.

Note that we are not suggesting the comparisons between MOPITT Level 3 product and aircraft measurements. Because the MOPITT Level 3 product is gridded data and represent the average value in a 1°×1° grid. Comparing the grid average value to an aircraft profile within it may be subject to large representativeness errors. Here we only show the sensitivity of agreement between MOPITT Level 2 data and aircraft profiles to the application SNR filter. This statement is also added to the section 4.4 of the manuscript. More discussion on the issue of representativeness errors can be found in section 5.

S38 – p16| 463-468: These 2 sentences are very late in the paper. They should be earlier, like in Section 2.2.
Response: We moved the two sentences to section 2.3 where we discuss the sub-grid variability and representativeness error in Figure 2.

S39 – p16| 468-476: This discussion on NO2 variability from GeoTASO to try to constrain CO variability seems irrelevant and late in the paper. I think the whole thing should be omitted.

Response: We think this discussion is highly relevant to the future direction of this study and other comparisons that have issues with urban variability for satellite spatial resolution. Nevertheless, we addressed the reviewer's comment by reducing the discussion on GeoTASO substantially to one sentence:

"*One possible way is to study $NO_2$ data retrieved from the Geostationary Trace Gas and Aerosol Sensor Optimization (GeoTASO) at very high resolution (250 m×250 m), to provide an upper estimate on CO variability*".

S40 – p17| 480: All the references to data should be split out into a "Data availability" section. See "Manuscript Composition" here: https://publications.copernicus.org/for_authors/manuscript_preparation.html.
Response: Thank you. We added a Data availability section and moved the relevant part from acknowledgement to it.

S41 – p17| 480: Best practice is for all data to be in a public repository. If this is not possible, then please provide contact information for how the aircraft data can be obtained including ARIAs and A-FORCE.
Response: We added contact info for ARIAs and A-FORCE in the Data availability section.

S42 – p17| 480: Include a last access date with all URLs.
Response: We included last access dates for all the URLs in the data availability section.

S43 – p17| 481-482: These seem like 2 references to the same MOPITT data? Which one should readers use?
Response: MOPITT data are available at both URLs. To avoid confusion, we deleted the second one "*https://earthdata.nasa.gov/ (Last access date: Jan 14th, 2020)*."

S44 – p17| 493: An "Author Contribution" section is needed: https://publications.copernicus.org/for_authors/manuscript_preparation.html.
Response: Thank you for pointing this out. We added an author contribution section after the data availability section.

S45 – p24 Table1: Could you please include the accuracy of CO from aircraft measurements somewhere?
Response: We added "Uncertainty" that includes precision and/or accuracy for each instrument in the Table 1.

S46 – p24| Table1: For better traceability, please list the CO scale the aircraft measurements were tied to (e.g., WMO-CO-X2004, WMOCO-X2014, WMO-CO-X2014A, CSIRO...).
Response: We agree this traceability is preferred, however, we were only able to confirm this for ARIAs (WMO-CO-X2014A), KORUS-AQ (WMO-CO-X2014A), SEAC[4]RS (WMO-CO-X2004), and DISCOVER-AQ MD, TX, and CA (WMO-CO-X2004). We added the available information to Table 1. However, the precisions/accuracies in Table 1 are based on the referenced literature for the aircraft measurements, which should be sufficient to describe the data we used.

S47 – p26: (No response requested) – Figure 1 is well done and has high information content

Response: Thank you.

S48 – p27| 776-777: I don't understand "vertical and horizontal variability" here. Does "horizontal" somehow correspond to how many km were flown? Or are these just the standard deviations of aircraft profiles. If so, then just say "are the standard deviations of the original aircraft observations."

Response: Thank you. We changed "*vertical and horizontal variability*" to "*the standard deviations of the original aircraft observations*".

S49 – p27| 777-778: I guess this is why the black and orange traces do not always match. Optionally consider plotting at layer centers (shifting up by about 50 hPa).

Response: Thank you. Ploting the $x_{rtv}, x_a, x_{in-situ}, and\ x_{transformed}$ on the surface, 900 hPa, 800 hPa etc layers is consistent with the naming in the main text as well as other figures and tables. We also have mentioned that "each MOPITT retrieval level corresponds to a uniform layer immediately above that level" in the figure caption as well as the main text.

S50 – p29| 793: Please define Delta log(VMR) here and explicitly include the base of the logarithm.

Response: Thank you. $\Delta\log_{10}$(VMR) is defined as $x_{rtv}$ - $x_a$ for MOPITT profiles and $x_{transformed}$-$x_a$ for the in-situ profiles. The use of $\Delta\log_{10}$(VMR) allows us to remove the impact of the a priori in the comparisons. We added this statement in the caption. We also added the base of the log (i.e., 10) here as well as in a few places in the main text.

S51 – p30| 803: Should "Figure 2" be "Figure 4"?

Response: Yes. Thank you for noticing this. We changed "Figure 2" to "Figure 4".

S52 – p31| 810: Numerically, what are considered "outliers"? Please also add to captions of Figures 8-11. Or just reference the caption the Figure 6 so it is less repetitive.

Response: An outlier is a value that is more than 1.5 times the interquartile range away from the top or bottom of the box. We added this statement in the caption of Figure 6, and referred to it in the caption of Figures 8-11.

S53 – p33| Figure8: Are these MOPITT biases compared with aircraft still? So the "200 hPa" values are the same as yellow values in Figure 6? Please specify or optionally consider showing as a % bias compared with the baseline "200 hPa" results.

Response: The biases are calculated against all (both urban and non-urban) in-situ profiles listed in Table 1. We have added this statement in the captions of Figures 8-11. In addition, the "200 hPa" values (gray) in Figure 8 are the same as yellow values (for all data) in Figure 6; the "100 km" values (gray) in Figure 9 are the same as yellow values (for all data) in Figure 6; the "12 h" values (gray) in Figure 10 are the same as yellow values (for all data) in Figure 6; and the "without SNR filter" values in Figure 11 are the same as yellow values (for all data) in Figure 6. We added this information in the corresponding captions too.

S54 – p33| Figure8: Please clarify that you are using "all" observations (both urban and non-urban).

Response: See reply to the comment # S53.

S55 – p34 Figure9: Optionally consider comparing against 100 km.
Response: See reply to the comment # S53.

S56 – p35 Figure10: Optionally consider comparing against 12 hours.
Response: See reply to the comment # S53.

S57 – Figure S3: It would help to remind readers that "in situ" is a combination of aircraft and models since values at 1050 hPa do not make much physical sense. (As a side observation it is interesting that MOPITT is so insensitive).
Response: We added "*in-situ profiles (combination of aircraft and reanalysis data as described in Section 2.3)*". in the caption of Figure S3.

S58 – Figure S5: Could you please provide more detail in the caption? Consider marking the level of highest aircraft measurements (presumably this is why there are straight lines).
Response: We extended the caption of Figure S5 to include more details:
"***Figure S5***. *Averaged in-situ profiles (combination of aircraft and reanalysis data as described in Section 2.3) under different assumptions of $P_{interp}$. For pressure levels below $P_{interp}$ (lower altitude), values are linearly interpolated using the highest-altitude aircraft measurement and reanalysis data at $P_{interp}$. For pressure levels above $P_{interp}$ (higher altitude), reanalysis data are used directly. For $P_{interp}$ equals 100 hPa, 200 hPa, 300 hPa, 400 hPa, and 500 hPa, the corresponding averaged in-situ profiles are shown by the blue, gray, yellow, green, and red lines, respectively. Taking the $P_{interp}$ equal to 100 hPa (blue line) as an example: for pressure levels below 100 hPa but above the highest-altitude aircraft measurement, the CO values are filled by linearly interpolation between CO values at the highest-altitude aircraft measurement and reanalysis data at 100 hPa; for levels above 100 hPa, CO values from reanalysis data are added to the in-situ profile directly.*"

Technical comments:

T1 - p1| 17: "The performance of the" could be omitted (and update has -> have)
Response: We have revised accordingly.

T2 - p1| 19-20: E.g., validate -> compare, using -> with
Response: We have revised accordingly.

T3 - p1| 25: suggest "performance" -> "agreement" and adding "with aircraft measurements" after V8T
Response: We have revised accordingly.

T4 - p2| 32: "allowed maximum" -> "maximum allowed" and "as criteria" -> "criterion"
Response: We have revised accordingly.

T5 - p2| 34-35: suggest "hence few MOPITT retrievals are included in the comparison."
Response: We have revised accordingly.

T6 - p2| 36: "overall smaller" -> "smaller overall"
Response: We have revised accordingly.

T7 - p2| 40: "retrievals that result for comparison." -> "retrievals for the comparison."
Response: We have revised accordingly.

T8 - p3| 58: "the most recently"
Response: We have revised accordingly.

T9 - p3| 83-84: suggest "...we compare MOPITT version...regions with aircraft profiles made over..."
Response: We have revised accordingly.

T10 - p3| 83: "version" is lowercase here but capitalized on p2| 55. Please be consistent throughout.
Response: Thank you. We changed "version" to be capitalized throughout the manuscript.

T11 - p4| 100: "retrievals" -> "observations"
Response: We have revised accordingly.

T12 - p4| 111: "The two" -> "These two"
Response: We have revised accordingly.

T13 - p5| 124: "determined" -> "considered"
Response: We have revised accordingly.

T14 - p5| 128-130: move/modify "the profiles over urban and non-urban areas are similar" to right after "We also notice..."
Response: The sentence "*We also notice for aircraft profiles sampled during KORUS-AQ, even though the averaged profile over urban regions has slightly higher CO concentration near the surface, the profiles over urban and non-urban are close.*" was revised to "*For aircraft profiles sampled during KORUS-AQ, the CO profiles over urban and non-urban regions are similar, even though the averaged profile over urban regions has slightly higher CO concentration near the surface.*"

T15 - p5| 139: omit "different instruments" (it's implied)
Response: We have revised accordingly.

T16 - p6| 148: "Only few" -> "Few"
Response: We have revised accordingly.

T17 - p6| 162: omit "below" (I initially was confused because I thought "below" meant lower pressure/higher altitude)
Response: Thank you for pointing this out. We omitted "below".

T18 - p6| 174-175: "We have investigated the..."
Response: We changed the sentence "*We have conducted further calculations to investigate the sensitivity of validation results to $P_{interp}$ in Section 4.1.*" to "*We investigate the sensitivity of validation results to $P_{interp}$ in Section 4.1.*"

T19 - p7| 179: "have been further" -> "are"
Response: We have revised accordingly.

T20 - p7| 186: "If fewer than five MOPITT retrievals are co-located with an in-situ profile, the..."
Response: We have revised accordingly.

T21 - p7| 187-191: I think it would be clearer if you reordered the description. a) In situ profile individually applied to AK and prior of each MOPITT retrieval to get xtranformed. b) xtransform averaged as log10. c) Corresponding MOPITT profile retrievals also averaged
Response: Thank you for the suggestion. We have rephrased these sentences "*If an in-situ profile is co-located with five or more MOPITT retrievals, these co-located MOPITT profiles are averaged as $log_{10}(VMR)$. These transformed in-situ profiles that are generated from the same in-situ profile are also averaged. Applying these corresponding different MOPITT a priori profiles and averaging kernels to the same in-situ profile results in different transformed in-situ profiles. These transformed in-situ profiles that are generated from the same in-situ profile are also averaged.*"

to

"*If an in-situ profile is co-located with five or more MOPITT retrievals (assume the number to be $N_{retrieval}$), then the following steps are used in the comparison with MOPITT: (a) the averaging kernel and a prior of each co-located MOPITT retrieval are applied to the in-situ profile (through equation 2) to obtain $N_{retrieval}$ of $x_{transformed}$. Note that applying these $N_{retrieval}$ sets of MOPITT a priori profiles and averaging kernels to the same in-situ profile results in differently transformed in-situ profiles; (b) the $N_{retrieval}$ of $x_{transformed}$ are averaged in $log_{10}(VMR)$ space; and (c) the $N_{retrieval}$ of MOPITT retrievals $x_{rtv}$ are also averaged.*"

T22 - p7| 200: "variability" -> "standard deviation"
Response: We have revised accordingly.

T23 - p7| 202 & 203: omit "retrieval" (the size does not depend on the retrieval algorithm, but is inherent in the MOPITT observation system)
Response: We have deleted "retrieval".

T24 - p8| 205: omit "very"
Response: We have deleted "very".

T25 - p8| 207: "validated" -> "compared with aircraft observations"
Response: We have revised accordingly.

T26 - p8| 209-210: "...show a sensitivity analysis in Section 4 to provide..."
Response: We have revised accordingly.

T27 - p8| 211: omit "validation"
Response: We have revised accordingly.

T28 - p8| 215: maybe "validation" -> "comparison with aircraft profiles"
Response: We changed the section title to "*3. MOPITT comparisons with aircraft profiles over urban and non-urban regions*"

T29 - p8| 224: "against observations" -> "against in situ observations"
Response: We have revised accordingly.

T30 - p9| 236: "Corresponding results" -> "These comparisons"
Response: We have revised accordingly.

T31 - p9| 244: "three levels" -> "three levels in Table 2"
Response: We have revised accordingly.

T32 – p9| 251: "...in terms of higher correlation coefficients and smaller biases..."
Response: Thank you. We have revised accordingly.

T33 - p9| 253: "provide" -> "evaluate", "evaluation against" -> "retrievals during", "campaigns" -> "campaigns with results"
Response: We have revised accordingly.

T34 - p9| 264: "in more favorable weather conditions" -> "during times with greater vertical mixing"
Response: We have revised accordingly.

T35 - p9| 271: "xin-situ, the" -> "xin-situ over non-urban areas, the"
Response: We have revised accordingly.

T36 - p9| 291: "concentrations all" -> "concentration for all"
Response: We have revised accordingly.

T37 - p9| 292: "For both" -> "For the higher 50% of measured mixing ratios both" and omit "if only the upper 50% of measured mixing ratios are considered"
Response: We have changed the sentence accordingly.

T38 - p12| 335-336: Suggest omitting "The validation results", and the second "validation" and changing "are" -> "does", "different for" -> change
Response: We have changed the sentence "*The validation results using 300, 400, or 500 hPa as $P_{interp}$ are not significantly different for the validation results against DISCOVER-AQ CA.*"
to
"*Using 300, 400, or 500 hPa as $P_{interp}$ does not significantly change the results against DISCOVER-AQ CA.*"

T39 - p12| 347: "the radius" -> "a radius"
Response: We have revised accordingly.

T40 - p12| 350: "close" -> "similar"
Response: We have revised accordingly.

T41 - p13| 356: "a smaller number of included" -> "including fewer"
Response: We have revised accordingly.

T42 - p13| 359: "a a more more" -> "a more"
Response: We have revised accordingly.

T43 - p13| 367: "especially" -> "including"
Response: We have revised accordingly.

T44 - p14| 399: "Level 3" -> "the Level 3"
Response: We have revised accordingly.

T45 - p14| 411: omit "process"
Response: We have revised accordingly.

T46 – p15| 423: suggest "overall" -> "mean"
Response: We have revised accordingly.

T47 – p15| 424: "to 3.5% for different levels"
Response: We have revised accordingly.

T48 – p15| 429: "to" -> "into"
Response: We have revised accordingly.

T49 – p15| 431: "compared with low"
Response: We have revised accordingly.

T50 – p15| 440: "as co-location criteria" -> "as a co-location criterion"
Response: We have revised accordingly.

T51 – p15| 441: "where a" -> "where only a"
Response: We have revised accordingly.

---

## Author Comment (AC2) · 31 Jan 2020

This manuscript by Wenfu Tang et al presented a comparison of the latest MOPITT CO V8 retrievals with aircraft measurements from DISCOVER-AQ, SEAC4RS, ARIAs, A-FORCE, and KORUS-AQ campaigns conducted over the US or East Asia. In addition, the sensitivities of validation results to assumptions and data filters applied during the comparisons of MOPITT retrievals and in-situ profiles were also performed and analyzed. The comparison between the MOPITT CO product with various version and the coincident observations has been previously performed by many scientists in many groups around the world. This study is an extension of previous study and the strategy for comparison has been used extensively in previous MOPITT evaluation and validation studies. However, this study is one of few studies that focus on comparison over around urban regions, this is interesting. Overall, this paper is well written and fits well within the scope of AMT. I recommend for publication though I rate the novelty of this paper as moderate. Since referee # 1 has listed numerous technical comments which are mostly overlapped with my comments. Here I don't present the repeated correction request.

Response: Thank you for your time and effort in reviewing our manuscript. We have addressed the comments accordingly. Please see below for details.

Extra minor revisions or comments are:

1. The Base map and color bar in Fig. 1 can be improved. It is hard to distinguish one from another. In latitude and longitude axis, the number like 30 should be 30°.

Response: We have changed colormap, color scale, and increased marker size. We also added the symbol for degree (°) in the latitude and longitude. See the updated Figure 1 in the manuscript for details.

2. What does the dashed line in Figs. 4 and 5 mean? The one to one lineïij§Should be stated in the caption.

Response: The dashed lines are one-to-one ratio lines. We added this information in the captions of Figures 4 and 5.

3. If you only compare the results at surface, 800 hPa, and 600 hPa. Then the expression should be the concentrations at surface, 800 hPa, and 600 hPa rather than the profiles at surface, 800 hPa, and 600 hPa.

Response: As described in the Section 2.3, we did compare the 10-level MOPITT profiles to 10-level in-situ profiles. Due to the lack of observations above 600 hPa, we only showed and discussed the results of comparisons below 600 hPa. The surface layer, 800-hPa layer, and the 600-hPa layer are selected to represent different conditions of the profiles below 600 hPa. Please also see the response to the comment # S20 from the reviewer 1. Nevertheless, we thank the reviewer for bringing this up, and changed the term "profile" to "concentration"/"value" when discussing a single layer. For example, we changed "*the overall agreements between values of MOPITT and in-situ profiles at the 800-hPa layer*" to "*the overall agreements between MOPITT and in-situ profiles at the 800-hPa layer*" in the section 4.1 to emphasize this statement is only for one layer.

4. Another confusing thing is that the MOPITT could have a very low DOFS at a given level with a limited range (Fig. 3). Thus, the retrieval should come more from a priori information rather than

the measurement. In other words, I guess, the good agreement between the MOPITT and aircraft at a given level is largely attributed to the a priori information and the smoothing effect in equation 2.

Response: The MOPITT V8N product does have a lower degree of freedom for signal compared to the MOPITT V8T and V8J products. Note that this manuscript mainly focuses on the V8T and V8J products (see the reply to the comment S4 of the reviewer 1). It is true that applying MOPITT AK and a priori (the smoothing effect in equation 2) to in-situ profile would reduce the difference between MOPITT profile and the in-situ profiles. However, this is the only correct way to perform such comparison. As stated by the MOPITT Version 8 Product User's Guide (available online at https://www2.acom.ucar.edu/sites/default/files/mopitt/v8_users_guide_201812.pdf), because of the dependence of MOPITT on the a priori information, users must transform these comparison datasets using the equation 2, so that the comparison data exhibit the same degree of smoothing and a priori dependence as the MOPITT product. We are aware of the impact of the a priori information in the retrievals. However, as described in Section 3.1, we explicitly removed the a priori information in the validation process following the method described in Deeter et al. (2017). Therefore, the good agreement between the MOPITT and aircraft at a given level is not largely attributed to the a priori information. In fact, the agreement would be much better than it is now if we did not remove the a priori information in the validation process.

---

## Author Comment (AC3) · 31 Jan 2020

General comments:
The aim of this paper is to evaluate two versions of MOPITT CO (V7 and V8) by comparison with aircraft observations from diverse campaigns all over the globe. Each version has two sub versions (V7-8T, V7-8N, V7-8J for thermal, NIR and TIR+NIR, respectively). Urban and non urban areas are the focus of the evaluation. This is a paper that complete the list of publications of the evaluation of the different versions of MOPITT CO. Lots of statistics are provided and the MOPITT users community could find some interest in order to interpret MOPITT data over urban areas.
Response: Thank you for your time and effort in reviewing our manuscript.

However, I found the comparison sometimes difficult to follow because of the large number of campaigns, the number of aircraft profiles by campaign, number of aircraft profiles over urban regions, the number of MOPITT CO profiles in different circles,.. Table 1 helps but if possible it would be nice to simplify in the text. Moreover, the title does not reflect totally the subject of the paper: the validation of the MOPITT CO retrievals is also over non urban regions. I suggest to change the title in that way.
Response: Please see the responses to the comments # G2 and S1 of the reviewer 1.

1) Moreover, the distinction of urban and non urban regions for the comparison of MOPITT CO with aircraft observations could mislead the reader. What is important in this study, is it the carbone monoxide emitted from the urban region or just the urban region with surface parameters different from non urban regions? Such surface parameters that are used in the retrievals of MOPITT CO (surface temperature, emissivity). At 600 hPa, some comparisons are done but this is above the boundary layer. There is a great chance that the CO measured by both MOPITT and the aircraft is transported from other regions that are not representative of urban regions. The author should clarify this point.
Response: We thank the reviewer for bringing this question up. The urban regions often have different surface parameters (e.g., surface temperature and emissivity), and usually but not always have higher CO concentrations than non-urban regions. However, the surface parameters are unlikely to impact the ultimate quality of MOPITT retrieval products (Pan et al., 1998; Ho et al., 2005). The goal of this study is to understand if MOPITT retrievals are able to represent conditions over urban regions given sampling, and cloud cover. In addition, the relatively large spatial and temporal variability of CO concentrations over urban regions makes the validation even more complex. Because of the complexity of urban regions and their connection with non-urban regions nearby, we also provide analysis at high CO concentrations regardless of landcover type. As the reviewer pointed out, the comparisons are done for the 600-hPa layer (usually in the free troposphere). It is possible that CO concentrations at this layer are transported from other regions that are not representative of urban regions. Even so, MOPITT retrievals at the 600-hPa layer are still impacted by the CO concentrations at other layers including the surface layer (equation 1). Therefore, the comparisons at 600 hPa is necessary. We have added the discussions above to the section 2.2. See the manuscript for details.

2) Also, it would be nice to have a clear recommandation on which MOPITT CO version to use. For example, after reading the table 2 of the paper, I found difficult to conclude on which version to use for urban or a non urban study as well. The statistics are often very similar and I was wondering what is the added value of V8 vs V7 and how significant the values are? It would be nice the authors discuss this point and conclude with clear recommendation in the conclusions on the use of the different versions of MOPITT CO.

Response: We thank the reviewer for pointing this out. The main goal of this study is not to compare MOPITT V8 and V7 products, but rather to validate the performance of MOPITT products over urban regions versus non-urban regions. The finding is that in general, MOPITT agrees reasonably well with the in-situ profiles over both urban and non-urban regions. As the reviewer pointed out, the statistics are often very similar, therefore we do not have recommendation for which version to use in terms of urban versus non-urban regions.

The MOPITT TIR-only, and TIR-NIR products both have their own advantages and disadvantages. MOPITT TIR-NIR products usually have higher DFSs and have enhanced the sensitivity to near-surface CO but may have larger retrieval noise compared to the TIR-only products (Deeter et al., 2011, 2013; Worden et al., 2010). The MOPITT V8 uses a new parameterized radiance bias correction method to minimize retrieval biases, therefore in general the MOPITT V8 performs better than V7 and is recommended (Deeter et al., 2019). A detailed description of MOPITT V8 products and their comparisons to MOPITT V7 products can be found in Deeter et al. (2019). We added the discussion below to the section 5 of the manuscript:

> "*The statistics are often very similar between different versions and products over urban and non-urban regions, and in general, MOPITT agrees reasonably well with the in-situ profiles in both cases. There is not, therefore, any reason to recommend the continued use of MOPITT versions earlier than V8 based on urban or non-urban region considerations. In general, MOPITT V8 is recommended (Deeter et al., 2019) as it uses a new parameterized radiance bias correction method to minimize retrieval biases, and has updated spectroscopic data for water vapor and nitrogen.*"

3) The Section 4.4 (Sensitivity to the signal-to-noise ratio (SNR) filters) is unclear to me. What are the conclusions we can draw from this section? Is level 3 useless? I didn't catch the point of this section. Maybe the authors could clarify on how to use Level 3 data over urban and non urban regions in the light of the use of such SNR filter.

Response: Please see the response to the comments # S37 of the reviewer 1.

Specific Comments:

Abstract:
In the paper, V7 and V8 of MOPITT CO are evaluated whereas only V8 is mentioned in the abstract.

Response: MOPITT V7 products is only used as a reference in the sub-Section 3.1 and is not the focus of this study. To avoid the confusion, we changed the sentence "*We focus on evaluating the recently released Version 8, as well as the Version 7, of the MOPITT TIR, NIR, and multispectral*

*TIR-NIR products.*" in the Section 2.1 (MOPITT retrievals and products) to "*We focus on validating the recently released Version 8 of the MOPITT TIR, NIR, and multispectral TIR-NIR products. We also include comparisons with the MOPITT Version 7 TIR, NIR, and multispectral TIR-NIR products in the Section 3.1 for reference.*"

**Section 3.3**

L 300-301: This means MOPITT CO concentrations are highly variable in circles where true concentrations are high. In this condition, what are the retrieval errors for these MOPITT pixels?
Response: We thank the reviewer for the question. We have conducted the calculation of the retrieval uncertainties, and added the statement below to section 3.3:
"*At higher 50% CO concentrations, the averaged retrieval uncertainties for the 600-hPa, 800-hPa, and surface layers, are 28%, 28%, and 29%, respectively. This is smaller than the averaged retrieval uncertainties at lower 50% CO concentrations (28%, 29%, and 30% for the 600-hPa, 800-hPa, and surface layers, respectively). We therefore conclude that the larger apparent biases at high CO concentrations are related to greater CO variability and representativeness error of the in-situ profile within the co-location radius used for analyzing the MOPITT data, rather than indicating larger retrieval uncertainties. Theoretically, MOPITT retrievals perform better with higher CO concentrations. The larger biases at high CO concentrations in Figure 7 implies that the relatively greater CO variability may overcome the impact of high CO concentrations. Addressing representativeness error and spatial variability in the comparisons between satellite and in-situ profiles is challenging, and will be discussed further in Section 5.*"

L 334: please correct the sentence
Response: The sentence is changed to "*At the 600-hPa layer, the agreements between the values of MOPITT and in-situ profiles are affected more by $P_{interp}$ compared to the those at the surface layer and the 800-hPa layer for comparisons with all the campaigns.*"

L 360: please correct the sentence
Response: We changed the sentence to "*We note that the usage of the largest radius (200 km) in this paper does not appear to degrade the overall results, even though representativeness errors generated from CO spatial and/or temporal variability are expected to increase. However, the use of the smallest radius (25 km) degrades the overall results by reducing the number of included MOPITT retrievals.*"

L 369: The sentence 'we note..' is unclear to me. Please clarify if necessary.
Response: We changed this sentence to
"*We note that when comparing to the ARIAs campaign, using 1h as the allowed maximum time difference decreases the biases at the surface layer, the 800-hPa layer, and the 600-hPa layer, compared to the cases using longer allowed maximum time difference (i.e., 3h, 6h, and 12h). This implies that the temporal variability is relatively large in the region.*"